

# Contrasting vertical distributions of recent planktic foraminifera off Indonesia during the southeast monsoon: implications for paleoceanographic reconstructions

Raúl Tapia[1], Sze Ling Ho[1], Hui-Yu Wang[1], Jeroen Groeneveld[2], and Mahyar Mohtadi[3]

[1]Institute of Oceanography, National Taiwan University, No. 1, Sec. 4, Roosevelt Road 10617 Taipei, Taiwan.
[2]Department of Geology, Hamburg University, D-20146 Hamburg, Germany
[3]MARUM – Center for Marine Environmental Sciences, University of Bremen, D-28334 Bremen, Germany

**Correspondence:** Raúl Tapia (raultapia@ntu.edu.tw), Sze Ling Ho (slingho@ntu.edu.tw)

**Abstract.** Planktic foraminifera are widely used in palaeoceanographic and paleoclimatic studies. The accuracy of such reconstructions depends on our understanding of the organisms' ecology. Here we report on field observations of planktic foraminiferal abundances (>150 $\mu$m) from 5 depth intervals between 0-500 m water depth at 37 sites in the eastern tropical Indian Ocean. The total planktic foraminiferal assemblage here comprises 29 morphospecies; with 11 morphospecies account-

ing for 90% of the total assemblage. Both species composition and dominance in the net samples are broadly consistent with those the published data from the corresponding surface sediments.

The abundance and vertical distribution of planktic foraminifera are low offshore west Sumatra, and increase towards offshore south Java and the Lesser Sunda Islands (LSI). Average living depth of *Trilobatus trilobus*, *Globigerinoides ruber*, and *Globigerina bulloides* increases eastward, while that of *Neogloboquadrina dutertrei*, *Pulleniatina obliquiloculata*,

and *Globorotalia menardii* remains constant. We interpret the overall zonal and vertical distribution patterns in planktic foraminiferal abundances as a response to the contrasting upper water column conditions during the southeast monsoon, i.e., oligotrophic and stratified offshore Sumatra (non-upwelling) vs. eutrophic and well-mixed offshore Java-LSI (upwelling).

Overall, the inferred habitat depths of selected planktic foraminifera species shows a good agreement with those from sediment trap samples and from surface sediments off Sumatra, but not with those from surface sediments off Java-LSI. The

discrepancy might stem from the different temporal coverage of these sample types. Our findings highlight the need to consider how foraminiferal assemblages and ecology vary on shorter timescales, i.e. from "snapshots" of the water column captured by plankton net to seasonal and interannual variability as recorded in sediment traps and how these changes are transferred and preserved in deep-sea sediments.

## 1 Introduction

Planktic foraminifera's diversity, community composition, population dynamics as well as their shell chemistry are sensitive to hydrographic parameters (e.g., temperature, salinity, food availability) of the upper ocean (Katz et al., 2010). Consequently,





their assemblage composition and the geochemical signature of their shells are routinely used to reconstruct past ocean conditions (Ding et al., 2006, 2013; Mohtadi et al., 2017; Steinke et al., 2014).

Since the early observations from Murray (1897), many studies have evaluated the relationship between seawater tempera-
ture and planktic foraminiferal distribution, and found increasing dominance of cold species with increasing latitudes (Bauer, 1976; Bé and Hamlin, 1967; Eguchi et al., 1999). This relationship with seawater temperature allowed the development of the first comprehensive mapping of past sea surface temperature (CLIMAP Project Members, 1976). Subsequent assemblage studies have attempted to improve the accuracy of past sea surface temperature (SST) reconstruction via the transfer function approach (e.g., Ortiz and Mix, 1997). Although seawater temperature has a large influence on the global distribution of planktic foraminifera (Kucera, 2007, 2009), other parameters such as salinity, oxygen content, food availability, turbidity, and upwelling intensity can also exert a strong control on the abundance, community structure, and vertical distribution of planktic foraminifera at regional scales (Anderson et al., 1979; Davis et al., 2021; Field, 2004; Lessa et al., 2020; Rebotim et al., 2017; Schiebel and Hemleben, 2017; Tolderlund et al., 1971).

In the last decades, geochemical proxies using the calcite tests of foraminifera (e.g., $\delta^{18}$O and trace elements such as Mg/Ca) have become a popular tool for the reconstruction of past ocean conditions (Katz et al., 2010; Lea, 2014; Schiebel et al., 2018). The robustness of paleoclimate reconstructions derived from foraminiferal calcite proxies is as good as our understanding of the multiple fields that planktic foraminiferal biology encompasses (Schiebel et al., 2018). For example, reconstructing past changes in the water column structure using multiple planktic foraminiferal species (e.g., surface vs. deep dweller) requires understanding of the ecology of the selected species, such as seasonality, habitat depth, and food requirements (Kemle-von Mücke and Oberhänsli, 1999; Kucera, 2007; Marchant et al., 2004; Schiebel et al., 2017). One common way of inferring planktic foraminifera habitat depth is by comparing the reconstructed parameters (typically Mg/Ca-SST) from surface sedi-
ments with instrumental data or climatological products (e.g., World Ocean Atlas) (Groeneveld and Chiessi, 2011; Hollstein et al., 2017; Mohtadi et al., 2011; Steinke et al., 2014; Tapia et al., 2015). However, this approach may be associated with uncer-
tainties arising from a myriad of processes during the settling, deposition, and burial that may lead to varying degrees of proxy signal alteration (Regenberg et al., 2014), selected proxy calibrations, and the instrumental database used for ground-truthing the proxy. "Snapshots" from the water column where planktic foraminifera dwell are thus providing additional information to improve proxy understanding.

The marginal seas of the Indonesian Archipelago are of great climatic importance. Here, the Indonesian Throughflow (ITF) connects the upper water masses of the Pacific and Indian oceans, exerting a strong effect on the salinity and heat exchange between these basins (Gordon and Fine, 1996; Gordon, 2005; Tillinger, 2011). This feature has fueled the scientific interest to better understand the role that heat exchange plays in climatic modulation on several time scales (Smith et al., 2020; Sprintall and Révelard, 2014). Consequently, a growing number of oceanographic reconstructions based on foraminiferal calcite have been published over the last decade to shed light on the changes in the regional oceanographic processes and climate (Dang et al., 2020; Ding et al., 2013; Fan et al., 2018; Karas et al., 2011; Mohtadi et al., 2014, 2017; Pang et al., 2021; Steinke et al., 2014; Wang et al., 2018). However, our understanding of the ecology of planktic foraminifera in this region, especially their vertical distribution, relies largely on indirect inferences based on surface sediments and sediment trap samples (Ding et al.,





2006; Mohtadi et al., 2007, 2009, 2011). Thus far there is only one field observation that is focused on the standing stock of
planktic foraminifera in surface ocean sampled using plankton net (Ujiié, 1968), but this study did not investigate the vertical
distribution of planktic foraminifera in the water column. To fill this gap, here we present depth stratified (0–500 m) plankton
net data from the Indonesian marginal seas off Sumatra, Java and the Lesser Sunda Islands (LSI) (Fig.1). The main goal of this
study is to shed light on the spatial distribution of planktic foraminifera during the southeast (SE) monsoon, on the relationship
between foraminiferal abundance and environmental parameters, and how these findings compare with sediment trap and core
top assemblage data to further improve our understanding of foraminifera-based proxy reconstructions in this region

### 1.1 Study area

Modern oceanography off Indonesia is strongly modulated by seasonal monsoons. During the SE monsoon from April to
October, the southeasterly winds from Australia induce Ekman pumping that generates upwelling along the coast of southern
Sumatra, Java and the LSI. The peak of upwelling-favorable winds occurs at the southernmost coast of Sumatra ($\sim$105º E)
during July-August, but the upwelling center moves northwestward reaching as far as 100º E and 2º S in October (Susanto
et al., 2001). Seasonal upwelling results in increased chlorophyll-*a* concentrations and reduced thickness of the depth of the
mixed layer (<20 m) south of $\sim$4º S (Fig. 1; Table A1 and A2). Furthermore, seasonal upwelling leads to a $\sim$2 ºC decrease in
SST in comparison to the non-upwelling season and a contrast in SST of $\sim$3 ºC between the northern and southern parts of
the study area. During the northwest (NW) monsoon from late October to early April, the wind direction is reversed, resulting
in downwelling, lower chlorophyll-*a* concentrations, thicker mixed layer, higher SST south of $\sim$4º S and a more uniform SST
distribution offshore southern Sumatra, Java and the LSI (Muskananfola et al., 2021; Qu et al., 2005; Susanto et al., 2001,
2006). At the interannual time-scale, the SST variability offshore southern Sumatra, Java and the LSI can be larger than 4 ºC,
showing the influence of climatic modes such as El Niño-Southern Oscillation (ENSO) or the Indian Ocean Dipole (IOD) (Qu
et al., 2005). During years of the El Niño (La Niña) phase of the ENSO and the positive (negative) phase of the IOD, intensified
(weakened) southeasterly winds result in stronger (weaker) coastal upwelling, leading to abnormally low (high) SSTs offshore
southern Sumatra and Java (Du and Zhang, 2015; Mohtadi et al., 2011; Qu et al., 2005).

## 2 Materials and methods

Multinet samples were collected between August and September 2005 during FS Sonne cruise SO-184 (Hebbeln and cruise
participants, 2006). The 47 sampling sites are grouped into seven land-sea transects off Sumatra (n = 4), Java (n = 2) and the
Lesser Sunda Islands (LSI) (n = 1) (Fig. 1; details in Table A1 and A2).

### 2.1 Collection of hydrographic data and foraminiferal samples

The characterization of the physicochemical properties of the water column (i.e., temperature, salinity, nutrients, oxygen,
and in situ chlorophyll-*a*) was obtained from CTD and water samples collected using a rosette water sampler equipped with
24 Niskin bottles (10-liter volume each) and a Seabird SBE911 probe (Hebbeln and cruise participants, 2006) across 45





water stations (see Table A1 and A2). These hydrographic data are available in the cruise report of SO184 (Hebbeln and cruise participants, 2006). Briefly, the dissolved oxygen was determined using an automated titrator (Titroline alpha), which

is controlled by a redox electrode and a color agent (starch solution), on a self-constructed titration board for ship cruises according to the WOCE protocol (Hebbeln and cruise participants, 2006). Chlorophyll-*a* measurement were performed using a SFM25 spectrofluorometer (KONTRON). The measurement were done with 1 cm cuvettes at a 435 nm excitation wavelength and a 667 nm emission wavelength (Hebbeln and cruise participants, 2006).

The plankton samples (n = 37; Table A1 and A2) were obtained using a MultiNet sampler (Hydro-Bios, Kiel, Germany).

The gear is comprised of five individual 64 $\mu$m nets with an opening of 0.25 m$^2$. The nets were lowered to a water depth of 500 m and vertically towed with a maximum winch speed of 0.2 m s$^{-1}$ along five depth intervals of 500 to 200 m, 200 to 100 m, 100 to 50 m, 50 to 25 and 25 to 0 m depth. At the end of each depth interval the MultiNet sampler was stopped to open the next net automatically closing the previous net. Once on board the samples were poisoned with 1 mL saturated HgCl solution and stored at 4 ºC (Hebbeln and cruise participants, 2006). The volume of water passed through the net opening was calculated

as the product of the height of the towed intervals and the area of the net opening.

## 2.2 Foraminiferal identification and census count

The taxonomic identification of the planktic foraminifera was based on Parker (1962), Kennett and Srinivasan (1983) and Hemleben et al. (1989). Here the species *Globigerinoides ruber* (white) can occur in two morphotypes (Mohtadi et al., 2009, 2011), namely *G. ruber sensu stricto* (*s.s.*) and *sensu lato* (*s.l.*). As *G. ruber* (*s.l.*) is now identified as a separate species,

*Globigerinoides elongatus* (Aurahs et al., 2009), we have included this distinction. The distinction was done according to the approach of Wang (2000); specimens with spherical chambers sitting on the previous suture and high arched primary aperture were classified as *G. ruber* (white), meanwhile, more compressed organisms with subspherical chambers and low arched primary aperture were classified as *G. elongatus*. In the case of the genus *Trilobatus*, we integrated the sacculifer-morphotype, i.e., individuals with a sac-like final chamber, into the counts of *T. trilobus* (individuals with a regular, globular

terminal chamber). The distinction of *Neogloboquadrina dutertrei* from *Neogloboquadrina incompta* was made according to the approach of Mohtadi et al. (2009). The samples were separated into four size classes (>500, 500–355, 355–250, and 250–150 $\mu$m) and stored in foraminiferal microslides. Here we report the foraminiferal abundances >150 $\mu$m as individuals per volume of seawater passing through the opening of the net (ind. m$^{-3}$).

Vertical habitat preferences were estimated using the total abundance of each species in each vertical profile. As we did not

discriminate between living and dead specimens, this approach may have led to an overestimation of the habitat depth, for instance when specimens were found in deep nets. Therefore, we refrain from interpreting individual multinet deployments, which may be prone to the aforementioned bias. Instead, we base our interpretations on the median value of transects, each consisting of at least three multinet deployments. The proportion of dead specimens, if any, likely varies across stations, thus averaging over stations yields estimates that are less prone to single-site bias. This approach might also mitigate potential biases

caused by patchy occurrences of planktic foraminifera in the water column, in both lateral and vertical directions (Meilland





et al., 2019). To facilitate comparison with previous studies and to avoid bias due to different sampling strategies, we used Average Living Depth (ALD) (Jorissen et al., 1995) to discuss the vertical habitat of planktic foraminifera. ALD is defined as:

$$ALD = \frac{\sum ni \cdot Di}{Ni}$$

Where ni is the number of specimens in the interval i of a particular species; Di is the midpoint of the sampled interval i;
and N is the total number of individuals for all the depth levels of that particular species. For the calculation of the ALD, we followed the approach suggested by Rebotim et al. (2017) by considering only stations with at least five individuals of a given species.

The 95% confidence interval of the species ALD for upwelling vs non-upwelling region (Table 1) was estimated using non-parametric randomization tests i.e., bootstrap (Manly,1997), as the distributions of ALDs of either region do not meet normal
distributions required for parametric tests due to small sample sizes. The upwelling region consists of transects 5–7, while the non-upwelling region consists of transects 1–3. We permuted the ALDs of each species for upwelling and non-upwelling region, respectively, simulating the differences in mean ALDs between these permuted sites with 999 replicates. We then sorted the sequence of 999 differences in mean ALDs, which approximated all possible outcomes of the lack of differences in mean ALDs between upwelling vs. non-upwelling sites. We determined the probability of the observed between-region difference in
mean ALDs as 1-quantile of the observed estimate in the sorted sequence. The probability (P-value) <0.05 represents significant differences in mean ALDs between regions. Statistical analysis was performed in R language, code available upon request.

## 3 Results

### 3.1 Hydrological conditions

Across the sampled depths (0–500 m) of the plankton nets, the water temperature ranges from ∼8 to ∼30 ºC, the salinity from
33.5 to 35.1 psu, the in situ chl-*a* from ∼0.2 to ∼0.7 mg m$^{-3}$ and the oxygen concentration from 1.2–∼5 ml L$^{-1}$(Fig. 2 a-c). Sea surface temperature shows a strong zonal contrast, with colder conditions off Java-LSI and warmer conditions off Sumatra (Fig. 2a). This zonal pattern remains across all the sampled depths from the surface to a water depth of 500 m (Fig.A1). Similarly, salinity also shows a strong zonal contrast, as fresher conditions dominate the upper 50 m of the water column off Sumatra while more saline conditions dominate the upper ocean off Java (<50 m) (Fig. 2b). This pattern reverses at depths >50 m, with
more saline condition off Sumatra than Java and the LSI.

In situ chl-*a* for 0–25 m water depth is on average ∼0.65 mg m$^{-3}$, with no zonal patterns across the study area (Table A1 and Fig. 2c). Lower-than-average chl-*a* values can be found at two oceanic stations from transect 3 off Sumatra, that is, sites GeoB10003 (0.28 mg m$^{-3}$) and GeoB10007 (0.34 mg m$^{-3}$) (Table A1). Despite the absence of a zonal divide in surface chl-*a*, at depths >50 m the vertical distribution differs between Sumatra and further east (∼105º E; Fig. 2c). Off Sumatra (<105º E),
chl-*a* is mainly restricted to the upper 100 m of the water column, while high chl-*a* values > 0.3 mg m$^{-3}$ (>105º E) can reach as deep as 500 m at the easternmost study area, i.e., Transect 7 around Sumba Island (Fig. 2c). The distribution of oxygen





content across the water column matches the distribution of in situ chl-*a*, i.e., high dissolved oxygen concentrations (>2 ml L$^{-1}$) are restricted to the upper 100 m of the water column off Sumatra, while high dissolved oxygen concentrations off Java and particularly around Sumba (Transect 7) can reach as deep as 500 m (Fig. 2d).

The thermal mixed layer depth (MLD$_{TEMP}$) for this region, defined as the depth where the temperature is >0.8 ℃ colder than the SST (Kara et al., 2000; Qu et al., 2005), roughly marks the top of the thermocline depth and ranges from ∼13 to ∼91 m (Table A1, A2 and Fig. 2a). The MLD$_{TEMP}$ varies zonally; off Sumatra the MLD$_{TEMP}$ (∼74 m) is on average twice the thickness of that off Java-LSI (∼33 m) (see Table A1 and A2). The barrier layer separates the well mixed upper ocean from the thermocline. Here, the barrier layer is defined as the MLD$_{TEMP}$ minus the mixed layer depth calculated using density, namely

MLD$_{DEN}$ (Qu and Meyers, 2005). It ranges between 0 to ∼72 meters of thickness and follows a similar spatial distribution as the MLD$_{TEMP}$, with an average thickness of ∼45 m off Sumatra and ∼2 m off Java-LSI (Table A1 and A2). The upper water column stratification, SI$_{0-200}$ is defined as the temperature difference between the sea surface and 200 m (Somavilla et al., 2017). The SI$_{0-200}$ values are higher off Sumatra than off Java and the LSI, indicating more stratified upper water column off Sumatra (Table A1 and A2). These observations suggest two contrasting hydrological conditions in the study area, with strong (weak)

water column stratification, thick (thin) mixed layer and barrier layer, and low (high) subsurface water entrainment towards the surface off Sumatra (off Java-LSI), in agreement with the observed geographical extension of the coastal upwelling (Fig. 1c) during the SE monsoon (Susanto and Marra, 2005; Susanto et al., 2001).

    Multivariate analysis (non-metric multidimensional scaling) performed on the in situ hydrographic data obtained during the cruise provides further evidence that the study area consists of three hydrologically distinct regions (Fig. 3), constituting of

transects 1–3, 4, and 5–7, respectively. The cluster off Java and the LSI that encompasses transects 5 to 7 is characterized by lower SST (<29 ℃), saltier surface ocean, shallow mixed layer, and chl-a with a larger vertical dispersion, i.e., conditions typical of seasonal upwelling during the southeast monsoon. Meanwhile, the cluster off Sumatra that encompasses transects 1 to 3 is characterized by higher SST (>29 ℃), fresher sea surface ocean, deep mixed layer, and chl-*a* with low vertical dispersion. A third cluster consists solely of Transect 4, which reflects transitional conditions as here we can observe a large

dispersion in the values of the parameters analyzed (see Table A1 and A2) and its location coincides with the northernmost extent of upwelling-favorable winds (see section 1.1). Based on the result of multivariate analysis and hydrographic data, we therefore categorize transects 1-3 as non-upwelling sites and transects 5-7 as upwelling sites.

### 3.1.1   Planktic foraminiferal assemblage and absolute abundance

The total composition of the planktic foraminiferal species comprises 29 morphospecies. Eleven morphospecies, namely *Glo-*

*bigerina bulloides*, *Globigerinella calida*, *Globigerinita glutinata*, *Globigerinoides ruber* (white), *Trilobatus trilobus*, *Globorotalia menardii*, *Neogloboquadrina dutertrei*, *Globorotalia hirsuta*, *Globigerina falconensis*, *Pulleniatina obliquiloculata*, and *Globigerinella siphonifera*, accounted for  90% of the total assemblage (Fig. 4a). The abundance of planktic foraminifera ranges from ∼3 to ∼80 ind. m$^{-3}$, and shows a strong zonal divide (Fig. 4b, c). The lowest abundances, with median values ranging between ∼4 and ∼12 ind. m$^{-3}$, occur off Sumatra (Fig. 4b, c) while, the highest abundances, with median values

ranging between 18 and 35 ind. m$^{-3}$, occur off southern Sumatra and Java-LSI (Fig. 4b, c).



## 3.2 Vertical distribution of palaeoceanographic-relevant species

Similar to the spatial pattern of total abundance in each transect (Fig. 4b), the vertical dispersion of planktic foraminifera abundance across the water column shows a strong zonal pattern (Fig. 5a). Off Sumatra, the vertical distribution of planktic foraminiferal abundance is characterized by a high concentration in the upper 50 m (i.e., >30% of the total abundance) and a

rapid decrease toward deeper levels with ≤15% of the total abundance occurring between 50 and 500 m water depth. Meanwhile, off Java and the LSI, planktic foraminifera can be found throughout the upper 500 m of the water column. Unlike for stations off Sumatra, here the first 50 m of water column are relatively poor in planktic foraminifera and the highest concentration (>20 % of the total abundance) occurs between 50 to 100 m water depth. This general pattern is true for all stations off Java-LSI except station GeoB10062-1 (Fig. 4c).

In the following section, we describe the vertical distribution of six species of planktic foraminifera with paleoceanographic relevance, namely *T. trilobus*, *G. ruber* (white), *G. bulloides*, *N. dutertrei*, *P. obliquiloculata*, and *G. menardii* (Caley et al., 2012; Ding et al., 2013; Mohtadi et al., 2017; Steinke et al., 2014; Tapia et al., 2019). *Trilobatus trilobus* exhibits a surface distribution with most of the organisms dwelling in the upper 50 m of the water column. Interestingly, some differences in their vertical distribution can be observed between sectors (i.e., Sumatra vs. Java-LSI) (Fig. 5b). In the Sumatra sector, *T. trilobus* is

highly concentrated within the upper 30 m of the water column, showing a small vertical dispersion, as its occurrence below 75 m is rare (Fig. 5b). Meanwhile, offshore Java and the LSI, *T. trilobus* shows a larger vertical dispersion with a relatively high concentration as deep as 100–200 m (Fig. 5b). The vertical distribution of *G. ruber* (white) shares some similarities with *T. trilobus*, that is, high concentration of specimens within the first 50 m of the water column off Sumatra and a larger dispersion off Java and the LSI (Fig. 5c). However, the vertical distribution of *G. ruber* (white) off Java suggests an even larger vertical

dispersion than that of *T. trilobus*, as *G. ruber* (white) shows that the lisocline of 20% of presence stretches from 100 to 400 m of water depth (Fig. 5c). *Globigerina bulloides* shows no clear pattern in its vertical preference off Sumatra, occupying both upper and lower depths. In contrast, off Java and the LSI this species seems to prefer water depths below the thermocline between 100 and 350 m (Fig. 5d). *Pulleniatina obliquiloculata* and *N. dutertrei* (Fig. 5e and g) show a similar distribution with high occurrence of individuals between ~50 and ~100 m water depth. This feature seems to be constant throughout the study

area. Although *G. menardii* is found mostly below 30 m water depth, its vertical distribution deepens along the study area, from dwelling at ~50 m water depth off Sumatra to ~150 m water depth off Java-LSI (Fig. 5f). Overall, abundance distribution of the selected species across the water column agrees with previous studies that categorize these species as surface-mixed layer dwellers (*T. trilobus*, and *G. ruber* (white)) or as deep-thermocline dwellers (*N. dutertrei*, *P. obliquiloculata*, and *G. menardii* (Birch et al., 2013; Faul, 2000; Hemleben et al., 1989; Lessa et al., 2020; Rebotim et al., 2017; Steph et al., 2009).

The ALD calculated from the total abundance (living + dead specimens) of the selected species ranges between 47 and 113 m water depth (Fig. 6). The ALD for *G. menardii*), *N. dutertrei*, and *P. obliquiloculata* indicates that the habitat depth of these species is located between ~67 to ~87 m. The ALD value for *G. bulloides* suggests that the habitat depth of this species in the study area is ~113 m, much deeper than the mixed layer in the study area. Meanwhile, the ALD for mixed-layer species



is ∼49 m for *T. trilobus* and ∼69 m for *G. ruber* (white), respectively. Interestingly, the ALD of surface-dwelling *G. ruber*
(white) is similar to that of some thermocline-dwelling species.

## 4 Discussion

### 4.1 Planktic foraminiferal abundance and assemblages off Indonesia

The total number of species (n = 29; size fraction >150 $\mu$m) observed in the plankton net samples collected during August and
September 2005 is higher than the number of species observed in the surface sediments off Indonesia (n = 18; size fraction >150
$\mu$m) (Ding et al., 2006) but lower than that observed in a sediment trap off Java (n = 37; size fraction >150 $\mu$m) (Mohtadi et al.,
2009). Higher diversity in our data compared to those of Ding et al. (2006) may be due to the loss of fragile and dissolution-
prone species (e.g., *Hastigerinella digitata*, *Turborotalita humilis*, *Globigerinita uvula*) in the sedimentary record, since only 4
of the 13 sites collected from the upwelling area off Java by Ding et al. (2006) are above the lysocline (∼2400–2800 m water
depth). On the other hand, the longer temporal interval spanned by sediment trap samples off Java (3 months) compared to our
sampling period (6 weeks) makes it possible to collect species with sporadic occurrence throughout the year.

The abundance of planktic foraminifera in the upwelling sector of Java-LSI is 4 to 8 times higher than that in the non-
upwelling-Sumatra sector (Fig. 4b). Higher foraminiferal abundance in the presence of upwelling is consistent with previous
studies, and is likely due to higher food availability when upwelling occurs (Kimoto, 2015; Schiebel and Hemleben, 2005;
Schiebel et al., 2001). Species such as *G. bulloides*, *G. glutinata* and *G. falconensis* have been classified as species associated
with upwelling conditions, and their abundances are positively correlated with the intensity of the upwelling (Brock et al., 1992;
Cayre et al., 1999; Conan and Brummer, 2000; Sautter and Sancetta, 1992). Species such as *N. dutertrei*, *P. obliquiloculata*,
and *G. menardii*  normally live in nutrient-rich waters below the mixed layer in the thermocline (Sautter and Thunell, 1991).
Meanwhile, *T. trilobus* and *G. ruber* (white) occur in areas with a thick mixed layer. These mixed-layer dwellers have a similar
life span (of two to four weeks), feeding strategies, and reproduction synchronized with the synodic lunar cycle (Schiebel
and Hemleben, 2017 and references therein). Compared to *T. trilobus*, which predominantly occurs under warm oligotrophic
conditions, *G. ruber* (white) displays a more opportunistic behavior, as this species can be abundant and occurs in a larger range
of trophic conditions, from oligotrophic to eutrophic due to its ability to feed on a greater variety of food sources (Schiebel and
Hemleben, 2017; Schiebel et al., 2018).

Off Sumatra without upwelling, *G. ruber* (white), *G. elongatus* and *T. trilobus* account for ∼44% of the total assemblage,
while the species associated with high productivity or upwelling conditions have a minor presence (*G. bulloides* (∼8%) and *G.
glutinata* (∼9%)) or are rare to absent ( *N. dutertrei*, <2%) (Fig. 7a). On the other hand, off Java and the LSI where upwelling
occurs, the assemblage composition is strongly dominated by *G. bulloides* (21%), *G. glutinata* (14%) and the deep-dwelling
species associated with high productivity i.e., *N. dutertrei*, *G. menardii*, and *P. obliquiloculata*, together contribute 18% to the
total assemblage. *Trilobatus trilobus* and *G. ruber* (white), contribute only 7% and 8%, respectively, to the total assemblage
off Java and the LSI (Fig. 7b). The spatial contrast in the composition and abundance of planktic foraminiferal species (Fig.
4 and 7) in upwelling vs. non-upwelling conditions suggests that the hydrographic changes related to upwelling govern the





distribution of planktic foraminifera. This finding is consistent with previous studies suggesting that off Indonesia, seasonal upwelling plays a critical role in modulating the ecology of planktic foraminifera (Ding et al., 2006; Mohtadi et al., 2007, 2009, 2011; Ujiié, 1968). Consequently, foraminiferal abundances (Fig. 4b) and species composition in plankton net samples

collected during the SE monsoon season reflect the transition, in space, from oligotrophic, deep mixed layer and more stratified upper water column conditions (Sumatra) to a more eutrophic, shallow mixed layer and well-mixed upper water column (Java and the LSI). The transition zone between the two hydrographic regimes for the period August-September is located off the Sunda Strait (∼103º – ∼105º W) (Fig. 1c) (Susanto et al., 2001). Altogether, the aforementioned observations suggest that changes in the temporal extent, intensity, and zonal coverage of the seasonal upwelling might have a profound effect on

the ecology of planktic foraminifera in the study area. Interestingly, the zonal divide in planktonic foraminiferal assemblage between non-upwelling and upwelling regions observed in our net data is also reflected to some degree in surface sediments (Fig. 7c and 7d). Similar to what we observed in the plankton net data of the SE monsoon, the assemblage in sediments off Sumatra is dominated by *G. ruber* while that off Java is dominated by *G. bulloides*. However, the proportion of less dominant species differs for plankton net and sediment samples. For instance, *G. calida* constitutes 12% of Java net samples but only

4% in the sediments. This discrepancy may reflect the different temporal coverage of these two sample types, i.e., surface sediments integrate over tens to hundreds of years whereas net samples provide only a "snap-shot" of the sampling period during the SE monsoon. The test of *G. calida* is relatively fragile, thus post-depositional processes like carbonate dissolution may also bias the assemblage in sediments (Ding et al., 2006). The effect of dissolution is likely not severe, as most of the stations are above the lysocline (Mohtadi et al., 2007), and some *G. calida* are found in sediments (Ding et al., 2006) albeit at

a lower proportion than in our plankton net data.

**4.2  Planktic foraminifera habitat depth off Indonesia**

Processes such as daily vertical migration and reproduction may also play a role in the vertical distribution of planktic foraminifera. The effect of daily vertical migration cannot be properly assessed by our sampling design, but there is strong evidence that argues against daily vertical migration in planktic foraminifera (Meilland et al., 2019). However, it cannot be

excluded that lateral patchiness of foraminiferal occurence affects the vertical distribution, as proposed recently by Meilland et al. (2019). To buffer against this potential caveat as well as potential bias due to dead specimens collected at depths (details in section 2.2), we interpret data averaged over several stations within each individual transects.

In the case of a modified vertical distribution due to synchronized reproduction – if some species reproduced consistently in phase with the full moon (Schiebel and Hemleben, 2017) – changes in the size class distribution should be noticeable before

and after the occurrence of full moon during the sampling period. Among all the species selected, *G. ruber* (white), *N. dutertrei* and *G. menardii* show changes in their size classes distribution consistent with this pattern (Fig. A1.2). The other species show equal distribution of the larger and smaller size fractions before and after full moon. Although we cannot completely rule out ontogenic vertical migration, the lack of coherence between the calculated ALDs and the moon phase argues against ontogenic vertical migration as primary driver of the habitat depth distribution in the study area.





The discussion in the following sub-sections focuses on species that are commonly used in geochemical analyses for pa-leoceanographic reconstruction, namely *G. ruber* (white), *T. trilobus*, *G. bulloides*, *N. dutertrei*, *P. obliquiloculata* and *G. menardii*. Importantly, the habitat depth of these species was also the focus of several previous studies in the region based on sediment trap and surface sediments (Mohtadi et al., 2007; 2009; 2011).

### 4.3    Dominant species in nutrient-poor waters: *G. ruber* and *T. trilobus*

Many studies have shown that *G. ruber* (white) and *T. trilobus* have mixed layer habitat preferences in oligotrophic conditions (Bé, 1977; Duplessy et al., 1981; Fairbanks et al., 1980; Kuroyanagi and Kawahata, 2004), thus they are considered reliable recorders of changes in the surface ocean at different timescales. However, the habitat depth of these species may change from area to area depending on the local hydrography, e.g., the depth of the mixed layer (Schiebel and Hemleben, 2017 and references therein).

The calculated habitat depth for *G. ruber* (white) (median = 69 m) (Fig. 6) is deeper than the habitat depth estimated from surface sediments for the study area (20 – 50 m) (Mohtadi et al., 2007, 2011). In contrast, the calculated habitat depth for *T. trilobus* (median = 49 m) (Fig. 7) is similar to the calcification depth estimated using surface sediments off Indonesia (∼50 m) (Mohtadi et al., 2011). The relatively great habitat depth shown by *G. ruber* (white) may be related to the lack of living planktic foraminifera specimens and the use of total counts in the calculation of the ALD (see section 2.2).

Previous studies have suggested that a deepening of the habitat depth due to the use of total counts can be considered marginal (Greco et al., 2019). For example, in the area influenced by the upwelling, including both live and dead (total) specimens in the calculation of ALD leads to an increase of only ∼4 to 15 m for *G. ruber* (Lessa et al., 2020; Rebotim et al., 2017). Similar overestimation, 2–13 m, in the habitat depth of *T. trilobus* is observed by including inclusion of dead specimens in the ALD, suggesting that the bias on ALD calculation caused by the inclusion of dead specimens should be comparable across mixed-

layer dwellers. If this is also true for our study area, the agreement in habitat depth of *T. trilobus* inferred from sediments and our ALD calculation indicates that dead specimens likely do not make up a large portion of the samples, thus the relatively deep ALD calculated for *G. ruber* (white) is likely not severely biased by the inclusion of dead specimens in the calculation.

### 4.4    Dominant species in nutrient-rich waters: *N. dutertrei*, *P. obliquiloculata*, *G. menardii* and *G. bulloides*

Species such as *N. dutertrei*, *P. obliquiloculata*, *G. menardii* and *G. bulloides* are normally associated with high food availability
and dwell in the upper water column across the mixed layer and upper part of the thermocline (Schiebel and Hemleben, 2017 and references therein). The habitat depth based on ALD calculation of the deep-dwelling species, *N. dutertrei* (median = 82 m) and *P. obliquiloculata* (median = 87 m) (Fig. 7) shows a good agreement with the habitat depth inferred from both sediment trap time series and surface sediments, i.e., 75–100 m for *N. dutertrei* and, 60–90 m for *P. obliquiloculata*, respectively (Mohtadi et al., 2009, 2011). These habitat depth estimates are close to the lower end of the range for these species in regions influenced

by the Benguela and Canary upwelling system in the Atlantic Ocean, that is, ALDs of 52 ±32 m for *N. dutertrei* and 45 ±31 m for *P. obliquiloculata*, respectively (Lessa et al., 2020; Rebotim et al., 2017). Similarly, the habitat depth of *G. menardii* off Indonesia is also at the lower end of its habitat depth observed off Africa in waters influenced by the Benguela Upwelling





System (ALD of 39 ±22 m) (Lessa et al., 2020). Notably, of all the species only *G. menardii* shows a shallower habitat depth (median = 67 m) (Fig. 8) than that inferred from sediment trap data (90 to 110 m) (Mohtadi et al., 2009). A habitat depth of

67–87 m water depth, for *G. menardii*, *N. dutertrei*, and *P. obliquiloculata*, places them just below the lower boundary of the mixed layer, which is on average 52 m (Table A1 and A2).

*Globigerina bulloides* has a median habitat depth of 113 m (Fig. 7); this value is almost twice the mean habitat depth estimated from surface sediments off Indonesia (∼50 m) (Mohtadi et al., 2011), and close to the low end of the observed habitat depth, i.e., 57 ±10 to 102 ±67 m, for this species in the Atlantic (areas under influenced of Benguela and Canary

upwelling) (Lessa et al., 2020; Rebotim et al., 2017). *Globigerina bulloides* is an opportunistic species whose abundance and habitat depth has been linked to food availability in the water column (Peeters and Brummer, 2002). Therefore, it is possible that its habitat depth follows the depths where food availability is the highest in the water column. The comparison of its median ALD value off Sumatra (non-upwelling) and off Java-LSI (upwelling) shows a deepening from ∼64 m to ∼152 m, respectively. This finding suggests that regional differences in the habitat depth of some planktic foraminifera might occur as a

result of seasonal upwelling.

### 4.5   Zonal differences in the habitat depth: non-upwelling vs. upwelling

Previous studies off Indonesia matching geochemical data ($\delta^{18}O_c$) from core tops and water profiles have suggested differences in the habitat depth of some planktic foraminifera species between the sectors of Sumatra, Java and the LSI (Mohtadi et al., 2007). Mohtadi et al. (2007) reported that off Sumatra, the *G.ruber*–$\delta^{18}O$ values reflected a habitat depth <50 m water depth,

while off Java and the LSI the *G.ruber*–$\delta^{18}O$ values were "out of range" (<0 m water depth). The authors speculated that the "out of range" geochemical signature off Java most likely reflects a greater contribution during the non-upwelling period when the water is warmer, fresher and more stratified. A similar situation was observed for *N. dutertrei*, as its geochemical signature suggests a deeper habitat depth in the non-upwelling Sumatra sector (50–75 m) relative to the upwelling Java sector (20–50 m) (Mohtadi et al., 2007). The notion that the seasonal upwelling off Indonesia may trigger changes in the habitat depth of planktic

foraminifera species is further supported by sediment trap data, wherein Mg/Ca and $\delta^{18}O$ data from planktic foraminifera (i.e., *G. ruber*, *N. dutertrei*, *G. menardii*, and *P. obliquiloculata*) vary with seasonal upwelling (Mohtadi et al., 2009). Therefore, the coherent, strong zonal shifts in the vertical dispersion of the selected planktic foraminifera in our plankton net samples (Fig. 5 a-g) and hydrographic parameters (i.e., SST, MLD, salinity, and chl-*a*) (Fig. 2a-c) in response to upwelling call for further scrutiny of potential zonal differences in the habitat depth values (Fig. 6).

The zonal disaggregation of the habitat depth (Fig. 8) into non-upwelling (transect 1 to 3) and upwelling (transect 5 to 7) sectors (see section 3.1) shows that *G. ruber* (white), *T. trilobus*, and *G. bulloides* have a greater habitat depth in the upwelling sector than the non-upwelling sector. There is a two-fold increase in their mean habitat depth (Fig. 8a-c) from non-upwelling to upwelling sector, i.e., from ∼33 m to ∼85 m for *T. trilobus*, ∼58 m to ∼97 m for *G. ruber* (white) and, ∼64 m to ∼152 m for *G. bulloides*. This finding, i.e., differing habitat depths in upwelling vs. non-upwelling regions for *T. trilobus* and *G. ruber*

(white), is further corroborated by the estimates of bootstrap 95% confidence interval and randomization test with replacement (Table 1; details of calculation in Method). Compared to other species, the statistical significance of the ALD difference in *G.*





*bulloides* is less strong (p = 0.057), likely due to the large spread in the vertical dispersion of this species. Although deeper than usually assumed for paleoceanographic reconstructions, the ALD values for *G. ruber* (white), *T. trilobus*, and *G. bulloides* in the upwelling sector are indeed within the ranges previously reported elsewhere for areas influenced by upwelling (Lessa
et al., 2020; Rebotim et al., 2017) (see section 4.2). A clear zonal divide can be observed, i.e., shallow ALD values occurring off Sumatra (non-upwelling) vs. deep ALD values occurring off Java-LSI (upwelling). Transect 7 in the LSI is in addition to monsoonal upwelling also under the influence of the ITF as it allows the passage of cooler and fresher water (Tillinger, 2011). The habitat depth of mixed layer dwellers here is not the deepest in the upwelling region, despite the low presence of mixed-layer dwellers (*G. ruber* (white) and *T. trilobus*) in the upper 50 m during the sampling period (Fig. 5), especially at stations
GeoB10065 and GeoB10070 (Fig. 4c). The ALD estimates of *G. ruber* (white) and *T. trilobus* at these two stations are the deepest along Transect 7 (the two lowest data points in the panel for *G. ruber* and *T. trilobus* in Figure 8). Station GeoB10070 is the easternmost site of the study, thus might be under a stronger influence of the ITF than the other stations along the transect. This is, however, not the case for station GeoB10065, which is flanked by several stations at which mixed layer dwellers are present in the upper 50 m. Therefore, it is possible that the vertical distribution at these two stations is not representative of
the transect nor of the influence of the ITF. Multivariate analysis also indicates that the hydrography at transect 7 is similar to that at transect 5 and 6, suggesting a negligible influence of ITF here. Although the calculated habitat depths off Sumatra (Fig. 8) of the surface dwellers show a relatively good fit with the estimated habitat depths based on geochemical data ($\delta^{18}O_c$ and, Mg/Ca temperatures) (Mohtadi et al., 2009, 2011), this is not the case for transects off Java and the LSI, where the overall lack of agreement between the sedimentary data and plankton net results is evident (Figure 8a, b and c).

370       The low abundance of *G. menardii*, *N. dutertrei* and *P. obliquiloculata* off Sumatra precludes ALD calculation, hence also the zonal comparison of their habitat depths (Fig. 8d-f) between Sumatra and Java-LSI. Despite their low abundance in net samples collected in August-September, these three species are found in relatively high abundances in surface sediments off Sumatra, constituting up to 13% of the assemblage (Fig.7c; Mohtadi et al., 2007). Thus, their occurrence offshore Sumatra might be temporally restricted to only the final part of the SE monsoon (October) when the upwelling center locked off the
Sunda Strait starts drifting westward (Susanto et al., 2001), triggering higher productivity offshore Sumatra. Alternatively, they might represent the positive IOD or El Niño years, when upwelling is generally stronger in the Eastern Indian Ocean and reaches further northerly latitudes. Off Java where their abundance is sufficiently high for ALD calculation, the data suggest that these three species share a similar niche at thermocline depths centered at ~90 m water depth, in agreement with the habitat depth inferred from surface sediments (Fig. 8d-f).

**4.6   Possible implications for palaeoceanographic reconstructions**

Field observations (plankton net and sediment trap data) provide insights into the modern ecology of planktonic foraminifera, as the habitat depth of some species is known to vary in time and at regional scale (Schiebel and Hemleben, 2017). Our plankton net data show that the habitat depth of mixed-layer dwelling *G. ruber* (white) and *T. trilobus* deepens in upwelling conditions, while thermocline-dwelling *N. dutertrei* and *P. obliquiloculata* thrive only in the upwelling region off Java and the LSI.





As with many zooplankton, the abundance of planktic foraminifera is linked to food availability, thus some species may change their habitat depth to maximize food acquisition. The habitat depth of symbiont-bearing species like *T. trilobus* and *G. ruber* is typically assumed to be restricted to the surface ocean or at least within the photic zone as they rely on their photosynthetic symbionts for nutrition. In addition, these species also catch prey and feed on a wide variety of food sources (Hemleben et al., 1989 and references therein), plausibly because the photosynthates produced by the symbionts are insufficient

to sustain the growth of the host (Bé et al., 1981; Caron et al., 1982). Indeed, a recent study showed that the nutritional contribution of the symbionts to the host is significantly smaller than that obtained by ingesting copepods, implying that the photosymbiosis in planktic foraminifera may not be the primary source of energy when preys are abundant (Takagi et al., 2018).

During the SE monsoon, the photic zone ranges between ∼50 m to ∼75 m, and the food availability off Java and the LSI

is high due to the enhanced predator-prey encounter fueled by the high phytoplanktonic biomass across the water column and upwelling-induced vertical mixing (Pécseli et al., 2014). Under this circumstance, it is plausible that omnivorous mixed-layer dwellers like *T. trilobus* and *G. ruber* (white) might adopt a feeding strategy that includes both photosymbiosis and preying, or even primarily the latter. Preying on other zooplanktons like copepods which have ontogenic and daily migration through water column means that planktic foraminifera are not limited to the photic zone for food. Moreover, copepods have been

found to shift to a deeper habitat as a strategy to maximize their retention within a coastal upwelling area (Peterson, 1998; Peterson et al., 1979; Verheye et al., 1991). In combination, the aforementioned factors may thus lead to a greater habitat depth for mixed-layer dwellers in the upwelling region off Java compared to the non-upwelling region off Sumatra.

The thermal gradient of mixed-layer and deep dwelling species (ΔT) is commonly used as a proxy for the thickness of the mixed layer and the position of the thermocline in the water column on glacial-interglacial time scales (Farmer et al., 2011;

Mohtadi et al., 2017; Steinke et al., 2014; Tapia et al., 2015). The ΔT calculated from the abundance-weighted temperatures of our plankton net data show that during the SE monsoon larger ΔT values occur off Sumatra than off Java-LSI where upwelling occurs (Fig. 9 and 10). Due to its relatively great habitat depth off Java-LSI, the abundance-weighted temperature derived from surface-dwelling *G. ruber* (white) and *T. trilobus* is comparable to that of thermocline-dwelling species, and substantially lower than that of their counterpart off Sumatra. Interestingly, smaller ΔT during upwelling conditions are also evident in the flux-

weighted data from a sediment trap off Java (Fig. 9 and 10), which show that the ΔT reduces from 4.3 ℃ during non-monsoon period to 1.2 ℃ during seasonal upwelling period (SE monsoon) (Mohtadi et al., 2011). Although both plankton net and sediment trap data are based on suspended or sinking foraminifera in the water column, they reflect a different signal in time and space – our net data reflect spatial difference during the sampling period spanning six weeks during the monsoonal upwelling season, while the trap data reflect temporal changes at one location that is under the influence of monsoonal upwelling. Despite

their inherent differences, both datasets, however, show that off Indonesia (ΔT) decreases as a function of upwelling dynamics. A more well-mixed upper water column in upwelling conditions, hence smaller surface-subsurface temperature difference, is also reasonable taking into account the weaker water column stratification during upwelling (Fig. 10). The latter is due to a thinner barrier layer and a shallower MLD$_{TEMP}$ (Fig. 2a and Table A1 and A2). This finding indicates that ΔT may be a





useful proxy for reconstructing past upwelling conditions off Indonesia, if the foraminifera produced during upwelling season
dominate foraminiferal test abundance in the sediments.

Mohtadi et al (2009) showed that ∼50% of the total annual foraminiferal flux off Java occurs during the SE monsoon season, and the fluxes during this season are largely centered around September, suggesting that our "snapshot" may be reasonably representative of the foraminiferal response to the prevailing ocean conditions during the SE monsoon off Indonesia. Geochemical data ($\delta^{18}O_c$ and Mg/Ca inferred temperatures) in marine sediments off Java show a broad agreement with the habitat
depth estimates from our plankton net data for subsurface dwellers (Fig. 8), but not for mixed-layer dwelling species. The situation differs off Sumatra, where the habitat depth estimates for both mixed layer and subsurface species derived from marine sediments and plankton net samples are in agreement, hence also the derived ($\Delta T$). Consequently, the zonal reconstruction of the $\Delta T$ based on surface sediments indicates a larger $\Delta T$ off Java-LSI than off Sumatra (Mohtadi et al., 2011), in contrast to that of our net data (Fig. 9 and 10). In other words, surface sediment data suggest larger $\Delta T$s in regions influenced by strong
seasonal upwelling.

This discrepancy between surface sediment and plankton net data off Java may stem from the different temporal intervals integrated by each sample type. The plankton net data reflect the conditions in the water column sampled during the SE monsoon, thus can be directly linked to the hydrographic processes that occurred during sampling. To some extent this is also true for sediment trap samples – limited temporal interval integrated by the samples at one location means that process
attribution can be better constrained. On the other hand, marine sediments integrate foraminiferal flux over tens to hundreds of years depending on the local sedimentation rate, and may be susceptible to post-depositional processes such as dissolution, bioturbation, and reworking which may alter proxy signal. Together, these issues make it challenging to quantify the relative importance of each processes and thus to ground-truth proxies. Nonetheless, paleoceanographic reconstructions are based on sedimentary material, which may have undergone the same post-depositional processes as the surface sediments. If the
relative importance of these processes would stay unchanged over time, then surface sediments are arguably the best modern analogue for paleoceanographic reconstruction. In this case, stronger upwelling off Indonesia is characterized by a larger ($\Delta T$) as indicated by the zonal pattern of surface sediments (Fig. 9 and 10). However, were the fluxes of planktic foraminifera and the post-depositional processes to change over time, it is within the realm of possibility that under some circumstances, e.g., strong positive IOD or El Niño years, the proxy signal produced in the water column during one-off events like upwelling
can be preserved in the sediments, especially if the resultant flux increases exponentially. Our findings highlight the need to consider multiple sample types to further constrain the analog used for downcore paleoceanographic reconstructions.

## 5 Conclusions

We examined the zonal and vertical distribution of planktic foraminifera off Indonesia during the boreal summer of 2005 at 37 stations. The stations were grouped into 7 land-sea transects off Sumatra and off Java-LSI. The factors driving the observed
distribution of foraminifer species were assessed using vertically resolved environmental data.



Twenty-nine species were identified that can be divided in two basic communities, i.e., one dominated by warm-oligotrophic-stratified water column species vs. cooler-eutrophic-well mixed water column species. Similarly, foraminiferal abundance shows contrasting distributions off Sumatra and off Java-LSI. The concurrent zonal shift in abundance and species composition of planktic foraminiferal and environmental parameters (SST, Chl-*a*, MLD and BL) in response to upwelling imply a close link between upwelling and the ecology of planktic foraminifera in this area. Similar to the abundances and species composition of planktic foraminifera, their vertical distribution across the water column shows a strong zonal differentiation i.e., shallow depths-low dispersion off Sumatra vs. deeper depths-larger dispersion off Java-LSI.

The calculated ALDs of the selected species are in broad agreement with typically assumed habitat depths in paleoceanographic reconstructions, with the exception of *G. ruber* (white) and *G. bulloides*. The possibility of an overestimation due to possible inclusion of dead specimens in the calculation cannot be entirely ruled out at this point, but several lines of evidence suggest that a severe overestimation is not likely. The species *G. ruber* (white) and *T. trilobus* show a strong deepening in their habitat depth off Java in comparison to the sector off Sumatra. This zonal divide is not evident in the habitat depth distribution of the subsurface dwellers *N. dutertrei*, *G. menardii*, and *P. obliquiloculata*, in part due to their low abundance off Sumatra. Compared to inferred habitat depth estimates based on surface sediments, those inferred from plankton net data show a better agreement for the subsurface species, while the relatively good agreement for the shallow dwellers is restricted to off Sumatra. Off Java and the LIS, surface-dwelling species in our net data have a much greater habitat depth than that derived from surface sediments. The discrepancy between plankton net and surface sediment data likely stems from the fact that each sample type integrates over different temporal duration, i.e., net samples reflect the conditions during the sampling period in SE monsoon while the marine sediments integrate over tens to hundreds of years of foraminiferal flux. Whilst each sample type has its pros and cons, our findings highlight the need to consider multiple sample types to further constrain the analog adopted for paleoceanographic reconstruction.

*Data availability.* Data generated in this study will be available on request to the main author until their online publication on PANGAEA (www.pangaea.de)

*Author contributions.* MM was responsible for the collection of planktic foraminifera and oceanographic data. RT analyzed the planktic foraminifera fauna. JG provided support for taxonomic identification. H-Y W contributed to the statistical analyses. The manuscript was written by RT and SLH with the contribution from all co-authors who approved its final version

*Competing interests.* The authors declare that they have no conflict of interest.





*Acknowledgements.* This study was funded by DFG grant HE3412/15-1 (MM), NTU Core Research Team fund 109L892604 and 110L890705 (SLH). RT acknowledges funding by MOST grants 108-2116-M-002-008, 109-2116-M-002-014 and 110-2116-M-002-007. We thank the
crew and scientists participating in the PABESIA Cruise (SO-184). We are grateful to T. Sagawa (U Kanazawa) and Y. Kubota (National Science Museum) for providing a copy of the paper by Ujiie (1968) stored at National Museum of Nature and Science in Tokyo, Japan.





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



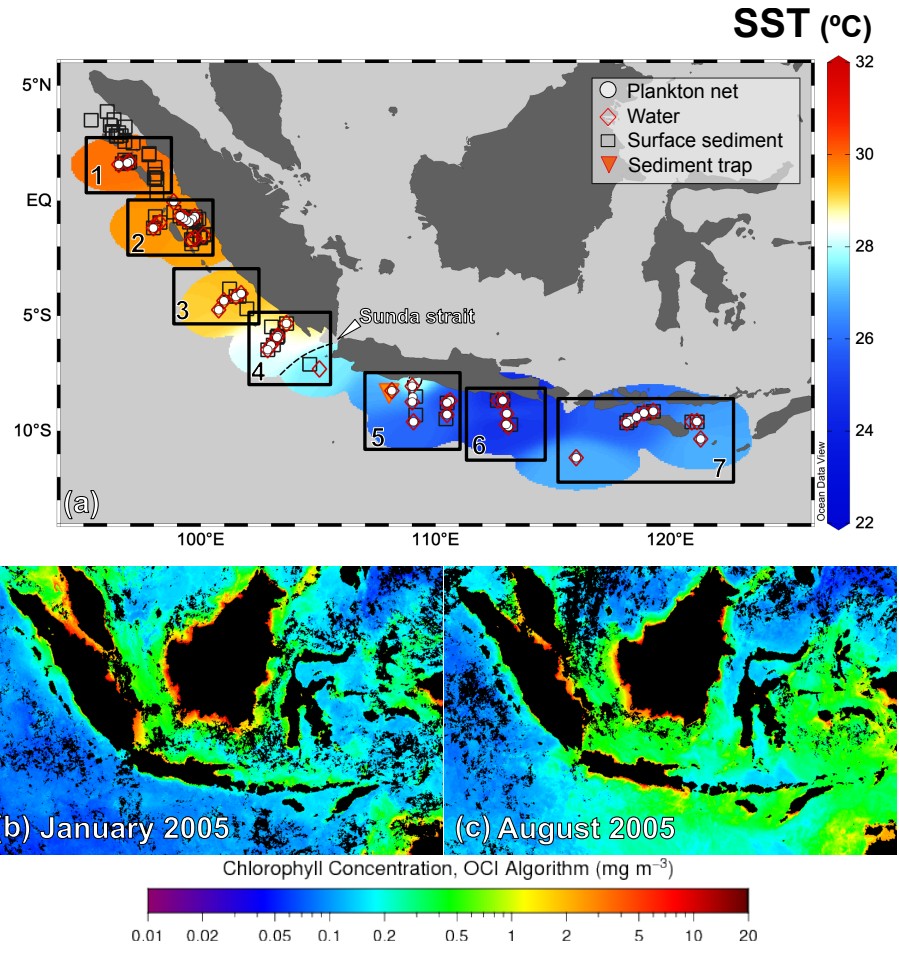

**Figure 1.** (a) Distribution of the in-situ sea surface temperature in the study area during PABESIA cruise on board of R/V Sonne in August 2005 (Hebbeln and cruise participants, 2006). The study area was divided into seven land-sea transects (numbered rectangles). The stations are divided into plankton net (white circles) and water (red diamond). The location of the sediment trap (Mohtadi et al., 2009) and surface sediments (Mohtadi et al., 2007) are indicated by triangle and squares, respectively. Upper ocean chlorophyll-*a* (Chl-*a*) (retrieved from https://oceancolor.gsfc.nasa.gov) during (b) January (Non-monsoon) and (c) August (SE-monsoon) 2005 i.e., the sampling period.



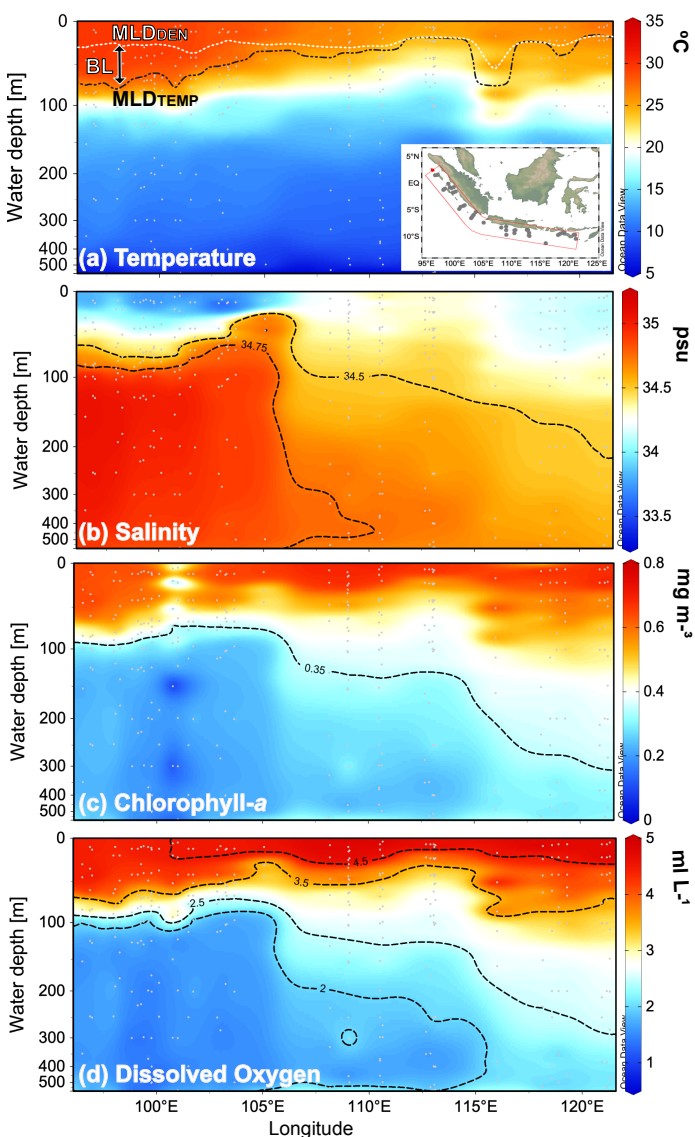

**Figure 2.** Cross sections of (a) water temperature, (b) salinity, (c) Chl-*a* and (d) oxygen content of the upper 600 m of the water column during the sampling period. Location of the stations comprising the cross section are depicted in the inset of panel (a). Thermal mixed layer depth (MLD$_{TEMP}$), density mixed layer depth (MLD$_{DEN}$) and barrier layer (BL) are marked in panel (a). Sampled depths depicted by gray squares (Hebbeln and cruise participants, 2006).



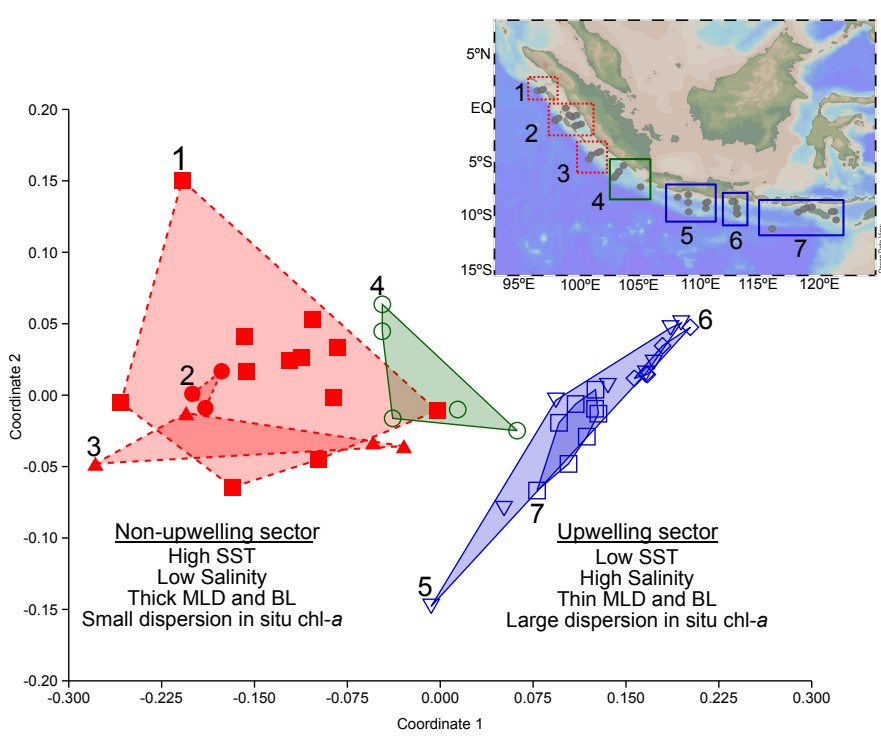

**Figure 3.** Non-metric multidimensional scaling of the in-situ hydrographic parameters shows three main clusters. Cluster 1 includes transects 5 to 7 off Java and the LSI influenced by upwelling (upwelling; blue). Cluster 3 includes transects 1 to 3 located off Sumatra which are not influenced by upwelling (non-upwelling; red). Cluster 2 consists of only transect 4 that represents a transitional condition between Sumatra and Java-LSI (green). Location of the stations are shown in the inset at the top right corner of the figure.



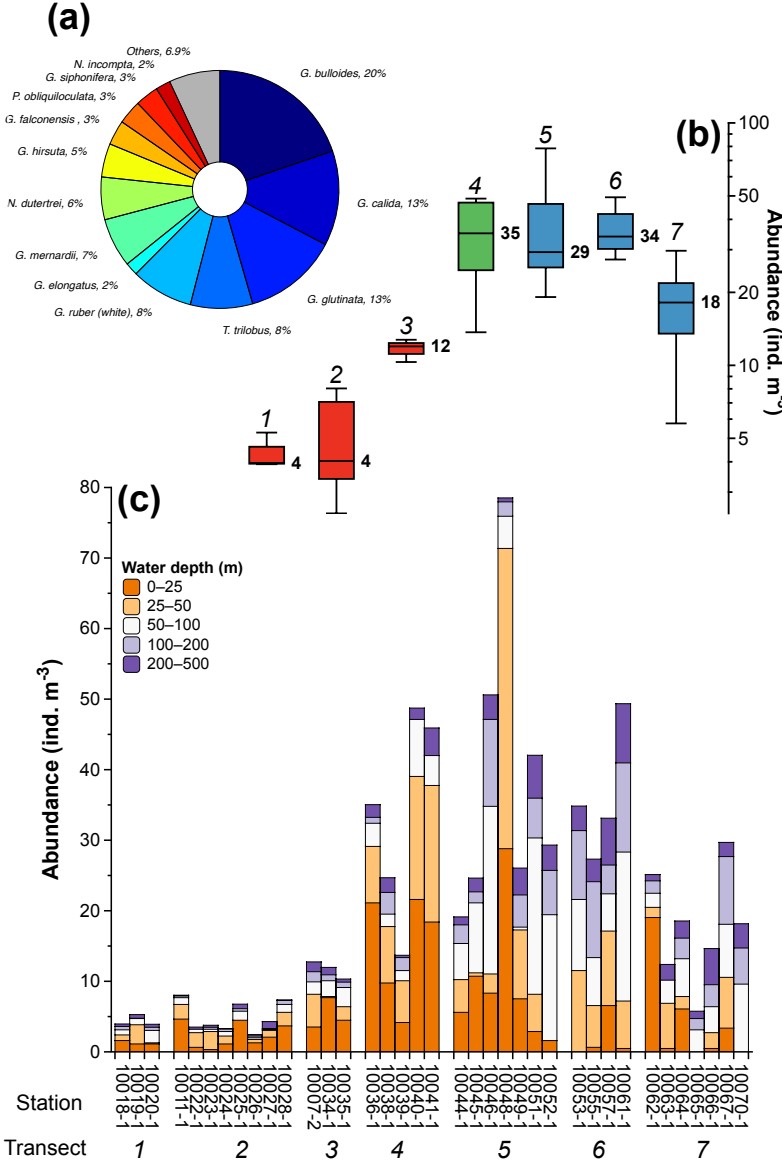

**Figure 4.** Summary of planktic foraminifera census counts off Indonesia. a) Species with contribution >2% to the total assemblage, b) abundance distribution of planktic foraminifera in transects 1 to 7 and, c) abundance of planktic foraminifera by water depth interval at each station. Numbers in (b) represent transects 1 to 7 (italic) and median values of transects (bold); colors in (b) depict sectors off Sumatra (red), transitional (green) and off Java-LSI (blue). The five sampled water depths are: 0–25 m (tangerine), 25–50 m (orange), 50–100 m (white), 100–200 m (lilac), and 200–500 m (purple). Stations are grouped by transect (number). Note the different scales for y-axis in b (log scale) and c (linear scale).

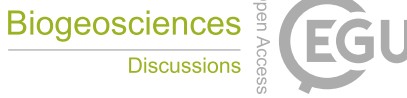

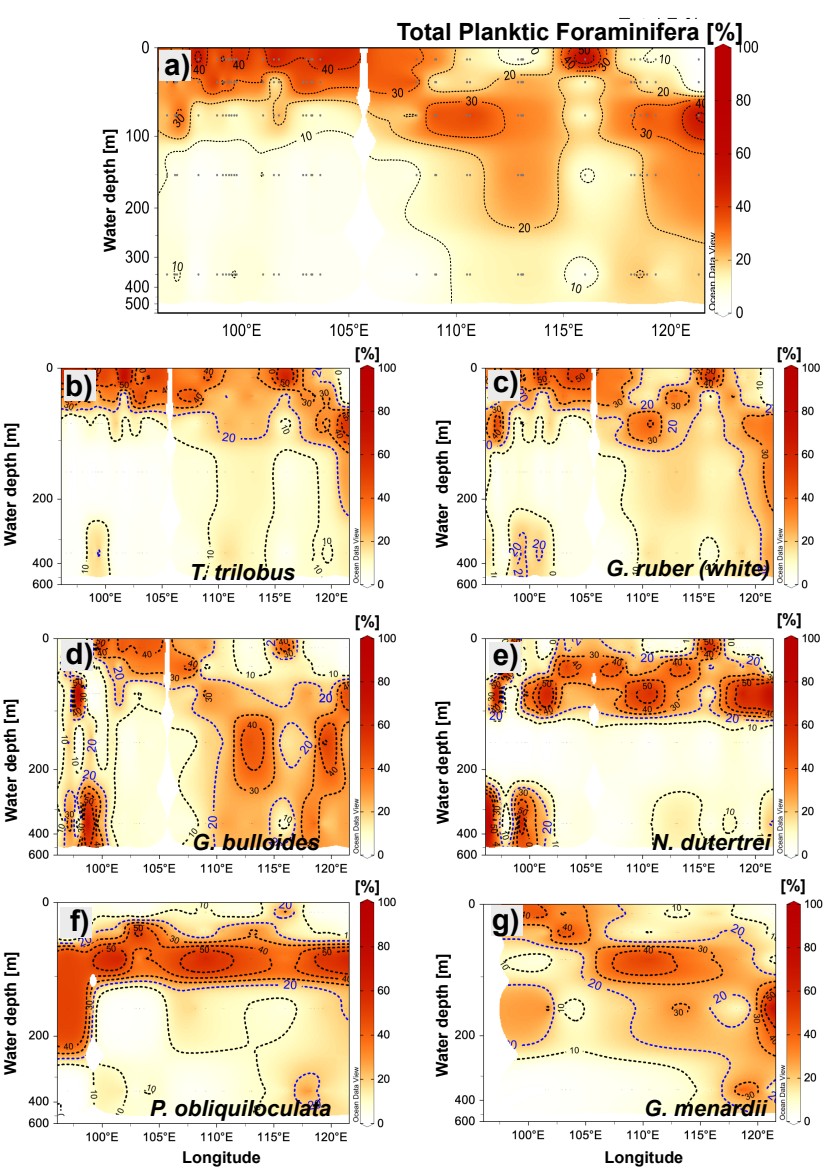

**Figure 5.** Cross sections showing the vertical distribution across the water column of a) the total community of planktic foraminifera and the species; b) *T. trilobus*, c) *G. ruber* (white), d) *G. bulloides*, e) *N. dutertrei*, f) *P. obliquiloculata*, and g) *G. menardii* .



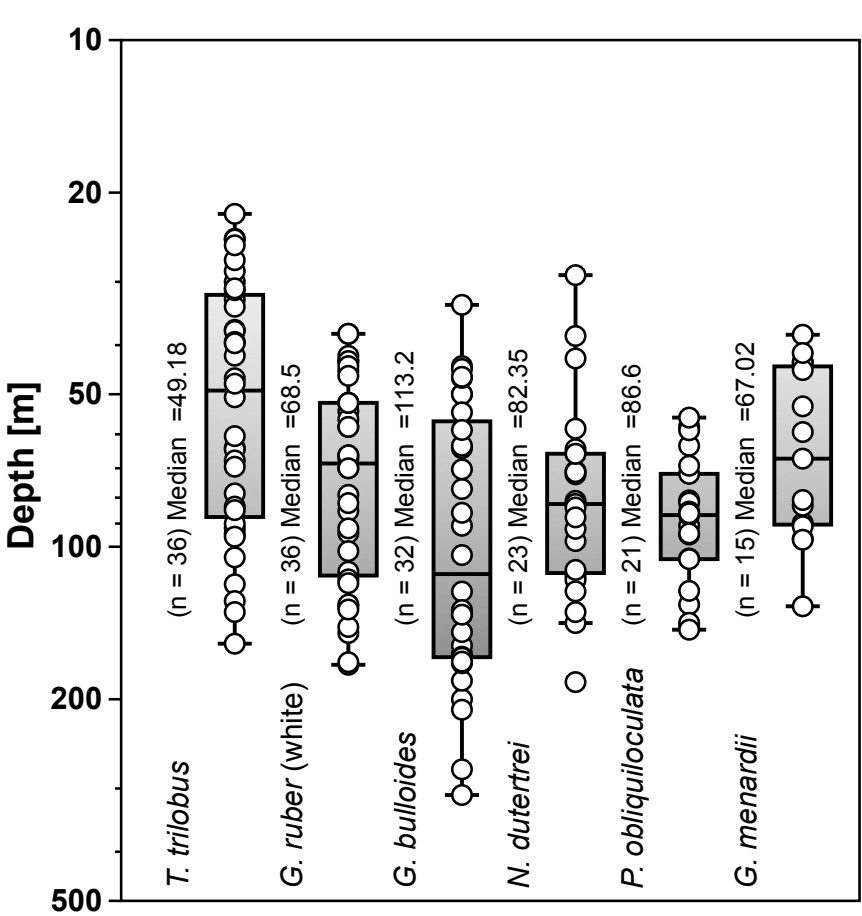

**Figure 6.** Average Living Depth (ALD) of the species *T. trilobus*, *G. ruber* (white), *G. bulloides*, *N. dutertrei*, *P. obliquiloculata* and *G. menardii*. The calculation includes only stations with more than 5 specimens per species (n). Note log scale for y-axis

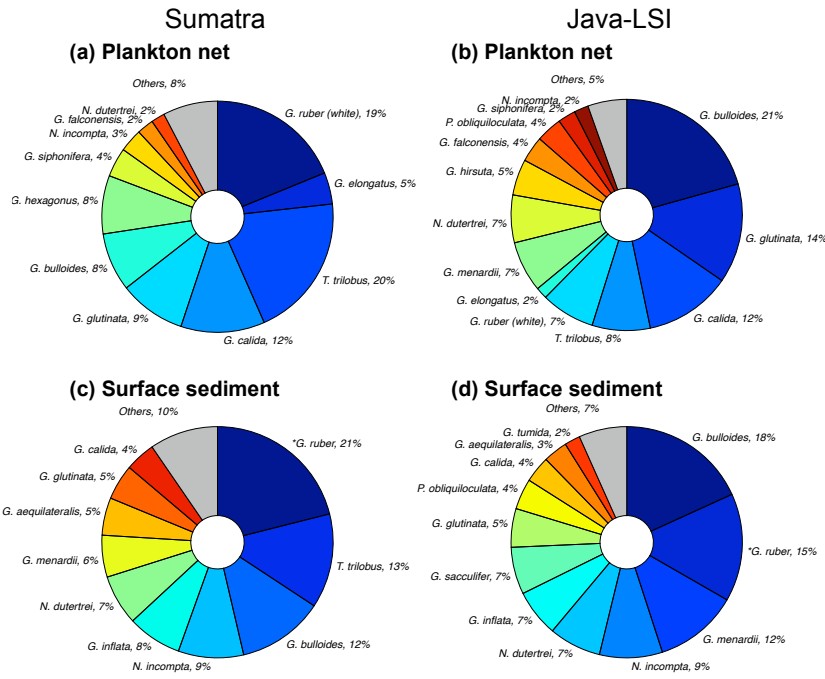

**Figure 7.** Planktic foraminiferal assemblage in (a and b) plankton nets (this study) and (c and d) surface sediments (data from Mohtadi et al., 2007) off Sumatra (Transect 1–3) and off Java-LSI (Transect 5–7) samples. Mohtadi et al. (2007) did not differentiate between morphotypes; *G. ruber* data comprise both morphotypes (*G. ruber* (white) + *G. elongatus*). Note that *G. ruber* (white) and *G. elongatus* data from plankton nets are plotted side by side to facilitate visual comparison with *G. ruber* in sediment data.

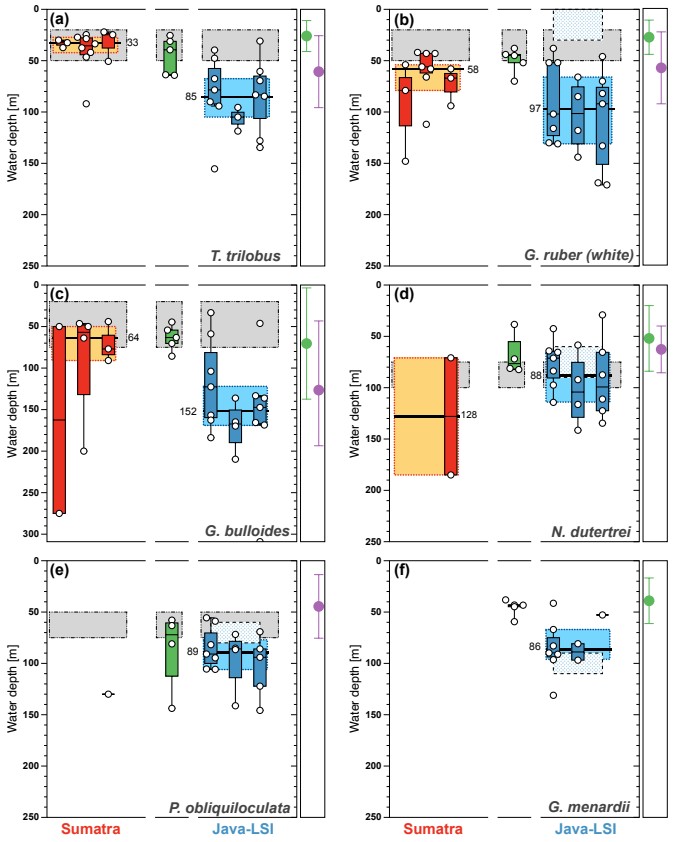

**Figure 8.** Average Living Depths (ALD) distribution for a) *T. trilobus*, b) *G. ruber* (white), c) *G. bulloides*, d) *N. dutertrei*, e) *P. obliquiloc-ulata*, and f) *G. menardii* per transect off Sumatra (red), transitional (green), and Java-LSI (blue). The calculation only considers stations with more than 5 counts per species. The yellow and light blue boxes depict the 1st and 3rd quartiles of the ALD values off Sumatra (non-upwelling) and off Java-LSI (upwelling); black bars and numbers depict the median value per sector. Boxes depict the inferred vertical distribution from surface sediments (gray) along Indonesia and from the sediment trap (pattern) off Java (Mohtadi et al., 2009; 2011). Colored dots and vertical lines depict calculated mean ALDs and vertical dispersion based on living foraminifera data from areas influenced by the Benguela (green) and Canary (purple) upwelling systems (Lessa et al., 2020; Rebotim et al., 2017). ALD for other species are presented in Appendices Fig. A.3



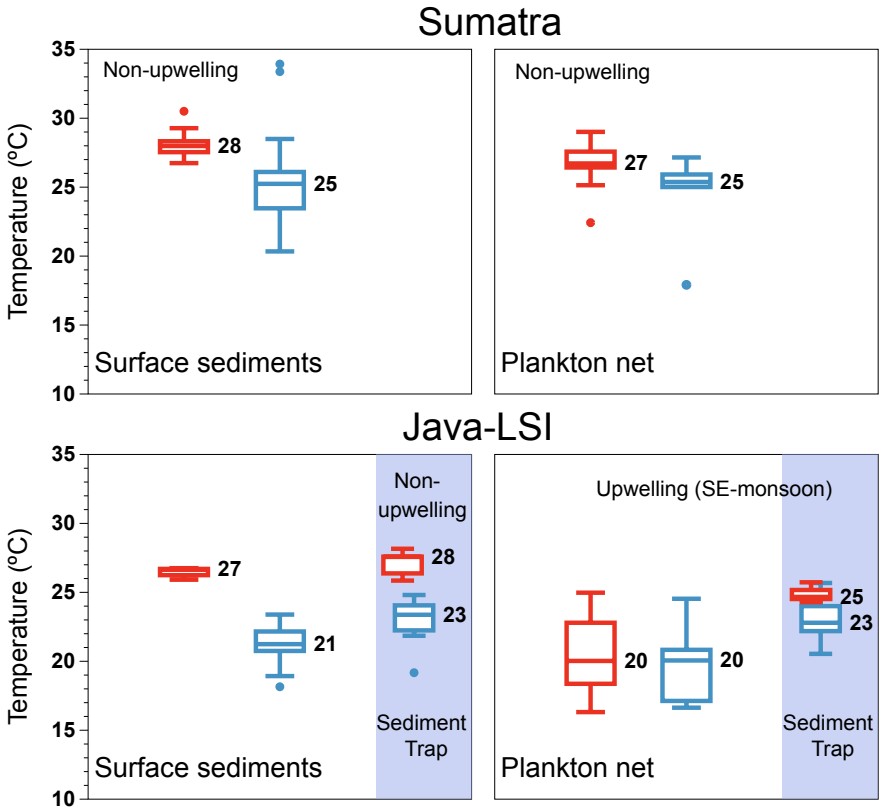

**Figure 9.** Comparison of the thermal gradient ($\Delta$T) off Indonesia inferred from surface sediments (Mohtadi et al., 2011), plankton net (this study) and from sediment trap (Mohtadi et al., 2009). During the SE monsoon the $\Delta$T calculated from the plankton net data shows that larger $\Delta$T values occur off Sumatra (non-upwelling) than off Java (upwelling) in agreement with the seasonal $\Delta$T off Java calculated from sediment trap (circles) with larger ($\Delta$T) occurring during the non-upwelling season than the upwelling season. The color represents the inferred averaged temperatures for the mixed layer (combining *G. ruber* and *T. trilobus*; red) and thermocline (combining *N. dutertrei* and *P. obliquiloculata*; blue), numbers represent median values.



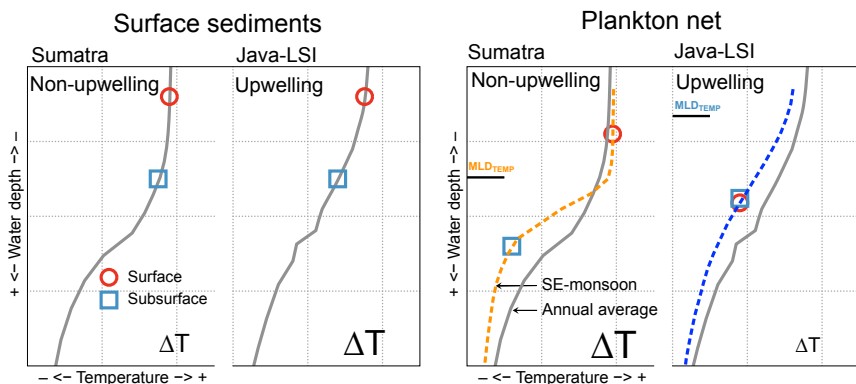

**Figure 10.** Schematic comparing the ΔT derived from mixed layer and thermocline-dwellers off Sumatra and Java-LSI, from a) surface sediment (inferred habitat depth) matching the mean annual conditions and b) plankton net (ALD) matching the water structure during the sampling period (SE-monsoon; August-September, 2005). Gray lines depict the average annual mean water structure offshore Sumatra and Java-LSI (band of 160 km wide; >200 stations from WOA2018; 0.25º) (Locarnini et al., 2018); dashed lines depict the average water column structure in each sector during the SE-monsoon based on in situ data collected during the PABESIA cruise (August-September, 2005); black line shows the average depth of the $MLD_{TEMP}$ (Table A1 and A2).





**Table 1.** 95% confidence interval of the habitat depth of 6 species of planktic foraminifera in upwelling vs. non-upwelling regions. Upwelling region consists of transects 5–7, while non-upwelling region consists of transects 1–3. Details of the calculation are described in section Method.

| Species | Non-Upwelling | | Upwelling | | | |
| --- | --- | --- | --- | --- | --- | --- |
| | Lower 95% CI | Upper 95% CI | Lower 95% CI | Upper 95% CI | Mean ALD $_{(upw - nupw)}$ | P-value |
| | (m) | (m) | (m) | (m) | (m) | |
| *T. trilobus* | 30.5 | 48.4 | 71.7 | 103.1 | 49.5 | <0.001* |
| *G. ruber*$_{(white)}$ | 56.5 | 87.1 | 82.4 | 118.4 | 29.1 | 0.017* |
| *G. bulloides* | 55.6 | 156.1 | 118.1 | 175.3 | 45.2 | 0.057 |
| *P.obliquiloculata* | NA | NA | 81.3 | 105.8 | NA | - |
| *N. dutertrei* | NA | NA | 72.9 | 102.1 | NA | - |

*Difference in ALD for the two regions is statistically significant. Abbreviation: CI for confidence interval; ALD for average living depth. Upwelling minus Non-upwelling (upw − nupw); NA for no data available



**Appendix A**

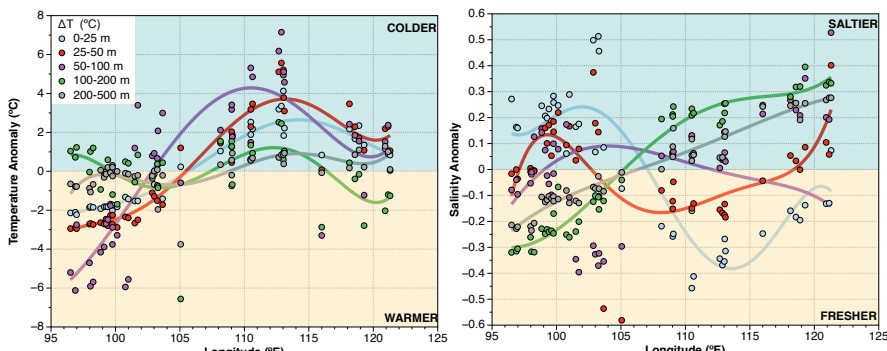

**Figure A1.** Longitudinal distribution of temperature and salinity anomalies across the upper 500 m of water column. Anomalies are calculated by subtracting the mean of all sites for a given depth interval (e.g., 0–25 m, 25–50 m, etc) from the data of a station for the same depth interval. Positive (negative) values are centered off Sumatra (off Java) showing strong zonal trend. The largest temperature change occurs at 50–100 m, roughly corresponding to the depth of the thermocline.



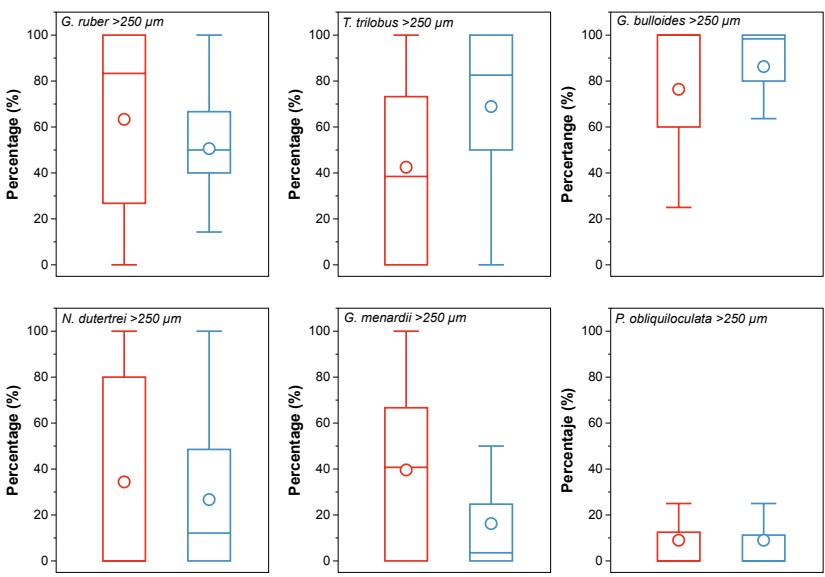

**Figure A2.** Size class distribution of the six selected species. *Globigerinoides ruber*, *N. dutertrei*, and *G. menardii* show a reduction in the proportion of the size class >250 $\mu$m after full moon (blue). The large difference in their median values suggest synchronized reproduction; *P. obliquiloculata* shows no differences in the size class distribution before (red) and after (blue) full moon; *Globigerina bulloides* and *T. trilobus* show an increase in the proportion of larger organisms after the full moon the 1st and 3rd quartiles; median values are shown by bar and circles represent mean values.



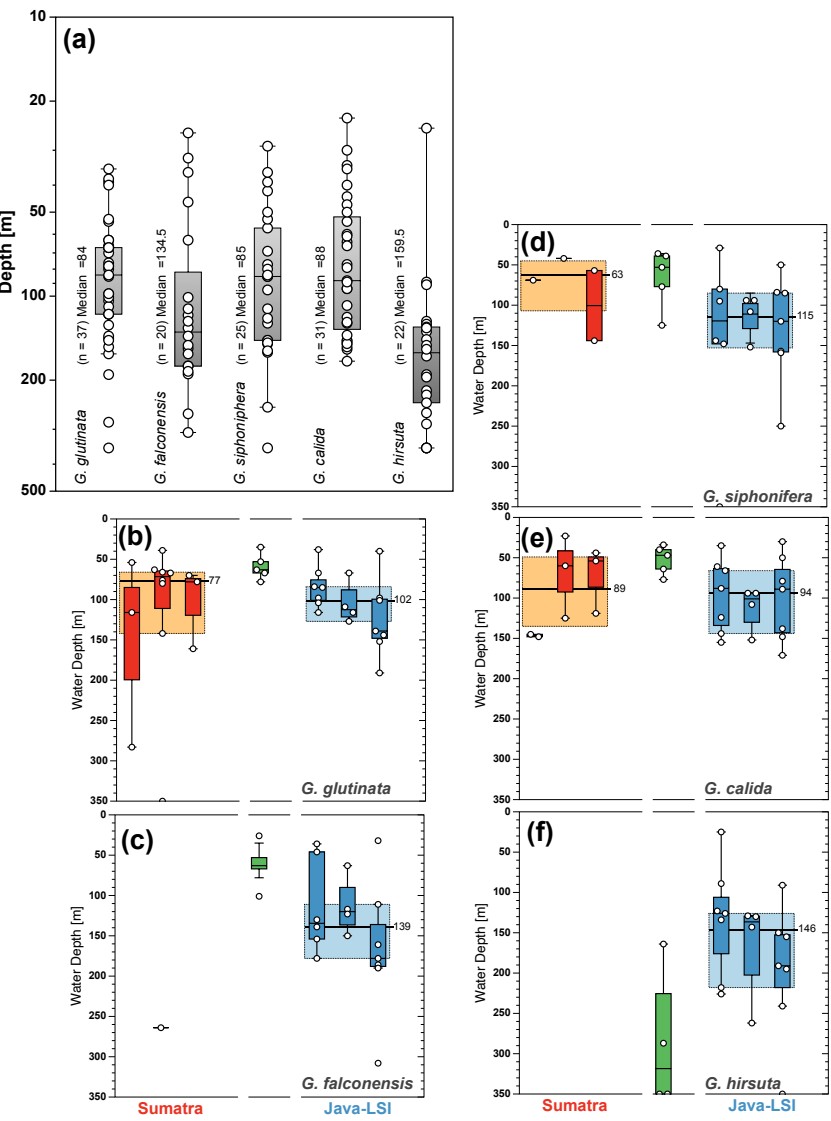

**Figure A3.** In addition to species with paleoceanographic relevance, we calculated the (a) ALD for the species *G. glutinata* (84 m), *G. falconensis* (139 m), *G. siphonifera* (85 m), *G. calida* (88 m), and *G. hirsuta* (160 m). The zonal comparison of these species (b-f) is only possible for three species; *G. calida* shows the smallest change in ALD (∼5 m) while *G. glutinata* and *G. siphonifera* show a change of 30 and 50 m in their habitat depth between sectors. The yellow and light blue boxes depict the 1st and 3rd quartiles of the ALD values off Sumatra (non-upwelling) and, off Java-LSI (upwelling). Black bars and numbers depict the median value per sector.





**Table A1.** Details of the plankton net stations (location, water depth, and sampling date) during cruise SO-184 (Hebbeln and cruise participants, 2006). The stations are divided into seven transects. Relevant environmental parameters during the sampling include the thermal mixed layer depth (MLD$_{TEMP}$), barrier layer thickness (BL), Stratification index (SI$_{0-200}$), observed sea surface temperature (SST), sea surface salinity (SSS), and in-situ chlorophyll-a (Chl-$a$) at the surface (0–25 m), and thermocline level (100–200 m) (Table 5.1 in Hebbeln and cruise participants, 2006). Sampling gear: Multinet (MN), Rosette Sampler (Rs), No data available (ND).

| Transect | Station GeoB | Lat. N | Long. E | Date | MLD$_{TEMP}$ m | BL m | SI$_{0-200}$ m | SSS psu | SST °C | Oxygen mL L-1 | Chl-a mg m$^{-3}$ 0-25 m | Chl-a mg m$^{-3}$ 100-200m | Gear | Depth m |
|---|---|---|---|---|---|---|---|---|---|---|---|---|---|---|
| 1 | 10018 | 1.570 | 96.512 | 08.08.05 | 83.6 | 51 | 17.6 | 33.7 | 29.9 | 4.9 | 0.646 | 0.220 | Mn, Rs | 2577 |
| | 10019 | 1.632 | 96.885 | 08.08.05 | 83.2 | 55 | 17.4 | 33.83 | 29.9 | 4.7 | 0.641 | 0.228 | Mn, Rs | 1465 |
| | 10020 | 1.678 | 96.980 | 09.08.05 | 77.8 | 52 | 17.5 | 33.83 | 30.0 | 4.7 | 0.641 | 0.241 | Mn, Rs | 1160 |
| 2 | 10011 | -1.191 | 97.986 | 06.08.05 | 82.3 | 70 | 16.8 | 34.0 | 29.6 | 4.4 | 0.634 | 0.253 | Mn, Rs | 3030 |
| | 10012 | -1.070 | 98.058 | 06.08.05 | 89.4 | 40 | 17.1 | 34.0 | 29.6 | 4.4 | 0.629 | 0.248 | Rs | 2096 |
| | 10013 | -0.958 | 98.266 | 07.08.05 | 76.0 | 30 | 17.0 | 34.0 | 29.6 | 4.3 | 0.619 | 0.212 | Rs | 927 |
| | 10022 | -0.051 | 98.850 | 10.08.05 | 71.5 | 52 | 17.1 | 33.8 | 29.7 | 4.4 | 0.639 | 0.227 | Mn, Rs | 707 |
| | 10023 | -0.857 | 99.407 | 11.08.05 | 70.4 | 43 | 17.4 | 33.8 | 29.6 | 4.4 | 0.635 | 0.204 | Mn, Rs | 1557 |
| | 10024 | -0.769 | 99.269 | 11.08.05 | 68.4 | 42 | 17.7 | 33.8 | 29.6 | 4.4 | 0.637 | 0.245 | Mn, Rs | 1384 |
| | 10025 | -0.675 | 99.123 | 11.08.05 | 67.9 | 34 | 17.3 | 33.8 | 29.6 | 4.6 | 0.654 | 0.245 | Mn, Rs | 1148 |
| | 10026 | -0.944 | 99.521 | 12.08.05 | ND | ND | ND | ND | ND | ND | ND | ND | Mn | 1636 |
| | 10027 | -0.809 | 99.653 | 12.08.05 | 64.4 | 52 | 17.6 | ND | ND | ND | ND | ND | Mn, Rs | 876 |
| | 10028 | -0.696 | 99.763 | 12.08.05 | 62.0 | 45 | 18.0 | 33.71 | 29.8 | 4.4 | 0.631 | 0.209 | Mn, Rs | 521 |
| | 10029 | -1.505 | 100.131 | 13.08.05 | 76.3 | 48 | 18.1 | 33.74 | 29.6 | 4.5 | 0.644 | 0.211 | Rs | 962 |
| | 10030 | -1.638 | 99.774 | 13.08.05 | 62.4 | 39 | 17.6 | 33.73 | 29.5 | 4.6 | 0.662 | 0.209 | Rs | 1757 |
| 3 | 10031 | -1.708 | 99.607 | 13.08.05 | 55.4 | 21 | 17.8 | 33.80 | 29.5 | 4.4 | 0.627 | 0.248 | Rs | 1661 |
| | 10003 | -4.751 | 100.767 | 03.08.05 | 90.9 | 72 | ND | 33.93 | 29.10 | 4.5 | 0.280 | 0.000 | Rs | 3176 |
| | 10007 | -4.354 | 100.996 | 04.08.05 | 85.8 | 54 | 16.8 | 33.70 | 29.2 | 4.5 | 0.338 | 0.239 | Mn, Rs | 598 |
| | 10034 | -4.165 | 101.499 | 15.08.05 | 62.8 | 21 | 16.6 | 33.83 | 29.3 | 4.6 | 0.654 | 0.205 | Mn, Rs | 992 |
| | 10035 | -4.036 | 101.733 | 15.08.05 | 66.7 | 25 | 16.9 | 33.82 | 29.3 | 4.5 | 0.652 | 0.228 | Mn, Rs | 997 |
| 4 | 10036 | -5.338 | 103.657 | 16.08.05 | 45.5 | 10 | 17.7 | 33.91 | 29.2 | 4.5 | 0.650 | 0.241 | Mn, Rs | 1498 |



**Table A2.** continued Table A1

| Transect | Station GeoB | Lat. N | Long. E | Date | MLD$_{TEMP}$ m | BL m | SI$_{0-200}$ m | SSS psu | SST °C | Oxygen mlL-1 | Chl-$a$ mg m$^{-3}$ 0-25 m | Chl-$a$ mg m$^{-3}$ 100-200 m | Gear | Depth m |
|---|---|---|---|---|---|---|---|---|---|---|---|---|---|---|
| **4** | 10038 | -5.937 | 103.245 | 17.08.05 | 51.2 | 39 | 16.5 | 33.48 | 28.50 | 4.52 | 0.651 | 0.227 | Mn, Rs | 1887 |
|  | 10039 | -5.867 | 103.294 | 17.08.05 | 53.9 | 36 | 16.7 | 33.54 | 28.56 | 4.50 | 0.646 | 0.246 | Mn, Rs | 1797 |
|  | 10040 | -6.475 | 102.857 | 18.08.05 | 62.0 | 25 | 16.5 | 33.49 | 28.00 | 4.57 | 0.657 | 0.236 | Mn, Rs | 2602 |
|  | 10041 | -6.274 | 103.008 | 18.08.05 | 51.9 | 19 | 16.4 | 33.92 | 28.45 | 4.50 | 0.647 | 0.232 | Mn, Rs | 1540 |
| **5** | 10043 | -7.310 | 105.062 | 19.08.05 | 34.1 | 3 | 16.8 | 34.07 | 27.55 | 4.47 | 0.642 | 0.398 | Rs | 2161 |
|  | 10044 | -8.055 | 109.015 | 22.08.05 | 33.5 | 2 | 16 | 33.93 | 28.53 | 5.39 | 0.775 | 0.301 | Mn, Rs | 3358 |
|  | 10045 | -8.743 | 109.020 | 23.08.05 | 40.3 | 6 | 15.5 | 34.25 | 26.52 | 4.72 | 0.677 | 0.326 | Mn, Rs | 3571 |
|  | 10046 | -9.604 | 109.063 | 24.08.05 | 50.2 | 1 | 14.9 | 34.24 | 25.90 | 4.81 | 0.691 | 0.376 | Mn, Rs | 2604 |
|  | 10048 | -8.255 | 108.147 | 25.08.05 | 43.2 | 0 | 15.8 | 34.21 | 26.74 | 4.78 | 0.687 | 0.335 | Mn, Rs | 3060 |
|  | 10049 | -8.785 | 110.496 | 26.08.05 | 37.7 | 1 | 14.5 | 34.45 | 25.24 | 4.70 | 0.676 | 0.335 | Mn, Rs | 1291 |
|  | 10051 | -9.293 | 110.497 | 27.08.05 | 31.7 | 5 | 14. | 34.15 | 25.89 | 4.78 | 0.687 | 0.364 | Mn, Rs | 2391 |
|  | 10052 | -8.694 | 110.634 | 28.08.05 | 35.9 | 6 | 14.9 | 34.40 | 25.60 | 4.88 | 0.701 | 0.350 | Mn, Rs | 1000 |
| **6** | 10053 | -8.677 | 112.872 | 29.08.05 | 23.7 | 0 | 14.6 | 34.36 | 24.43 | 4.17 | 0.600 | 0.321 | Mn, Rs | 1378 |
|  | 10054 | -8.681 | 112.668 | 29.08.05 | 19.0 | 1 | 13.7 | 34.34 | 24.59 | 4.16 | 0.599 | 0.314 | Rs | 1069 |
|  | 10055 | -9.248 | 113.050 | 30.08.05 | 23.8 | 0 | 14.9 | 34.34 | 24.59 | 4.16 | 0.599 | 0.314 | Mn, Rs | 2615 |
|  | 10057 | -9.822 | 113.107 | 31.08.05 | 26.3 | 1 | 14.3 | 34.26 | 25.94 | 4.76 | 0.684 | 0.348 | Mn, Rs | 1615 |
|  | 10061 | -9.729 | 113.024 | 02.09.05 | 12.7 | 2 | 14.3 | 34.35 | 23.65 | 4.29 | 0.617 | 0.354 | Mn, Rs | 2174 |
| **7** | 10062 | -11.166 | 115.999 | 03.09.05 | 76.4 | 1 | 12.6 | 34.24 | 26.94 | 4.67 | 0.671 | 0.390 | Mn, Rs | 5851 |
|  | 10063 | -9.646 | 118.149 | 04.09.05 | 24.4 | 0 | 13.8 | 34.15 | 25.61 | 4.78 | 0.687 | 0.385 | Mn, Rs | 2498 |
|  | 10064 | -9.539 | 118.304 | 04.09.05 | ND | ND | ND | ND | ND | ND | ND | ND | Mn | 2033 |
|  | 10065 | -9.223 | 118.894 | 05.09.05 | 56.4 | 3 | 13.4 | 34.19 | 25.43 | 4.49 | 0.645 | 0.409 | Mn, Rs | 1286 |
|  | 10066 | -9.394 | 118.575 | 05.09.05 | 29.5 | 4 | 13.6 | 34.18 | 26.11 | 4.80 | 0.690 | 0.412 | Mn, Rs | 1630 |
|  | 10067 | -9.149 | 119.290 | 06.09.05 | 38.0 | 9 | 14.6 | 34.13 | 26.43 | 4.75 | 0.682 | 0.441 | Mn, Rs | 1136 |
|  | 10068 | -9.595 | 121.152 | 07.09.05 | 16.0 | 2 | 15.4 | 34.12 | 26.87 | 4.71 | 0.677 | 0.428 | Rs | 2011 |
|  | 10069 | -9.608 | 120.921 | 07.09.05 | 14.2 | 2 | 14.2 | 34.12 | 26.62 | 4.77 | 0.686 | 0.422 | Rs | 1264 |
|  | 10070 | -10.359 | 121.303 | 08.09.05 | 22.2 | 0 | 13.6 | 34.13 | 26.94 | 4.82 | 0.693 | 0.406 | Mn, Rs | 1509 |