# Peer review of "Contrasting vertical distributions of recent planktic foraminifera off Indonesia during the southeast monsoon: implications for paleoceanographic reconstructions"

_Biogeosciences, 2021_

## Author Response (AR1)

**Dear Editor,**

Here, we would like to re-submit the revised version of the manuscript titled "Contrasting vertical distributions of recent planktic foraminifera off Indonesia during the southeast monsoon: implications for paleoceanographic reconstructions"

We would like to thank the editor and the two anonymous reviewers for their helpful comments and suggestions. We have made changes to our manuscript accordingly by adding more discussion and rephrasing the text that is considered unclear by the reviewers. We have included additional references and modified Figures 1, 2, 4, 5, 6, 8, 9, and 10 as suggested by the reviewers.

A list of all the changes made can be found in the point-by-point response to the reviewers' comments. We submit two versions of the revised manuscript, i.e., with and without tracked changes. We refer to the line number in our rebuttal letter as viewed with tracked changes. For your convenience, the reviewers' comments are in black while our responses are in blue.

We hope that we have successfully addressed all the points raised and that the improved manuscript now meets the standards of Biogeosciences.

Thank you very much for handling this manuscript.

Yours sincerely,
Raúl Tapia

**Lines number refer to the track changed version**

**Reviewer #1**

The contribution "Contrasting vertical distribution of recent planktic foraminifera off Indonesia during the southeast monsoon: implications for paleoceanographic reconstructions" presents new plankton net data from the eastern tropical Indian Ocean. They analyzed the planktic foraminiferal abundances from 5 depth intervals between 0-500 m water depth at 37 sites covering the Indonesian marginal seas off Sumatra, Java and the Lesser Sunda Islands in order to shed light on the spatial distribution of planktic foraminifera during the southeast monsoon and established the relationship between their abundance and the environmental parameters to finally compare those findings with sediment trap and core top assemblage data to further improve the foraminifera-based proxy reconstructions in the region. This study represents an important contribution to the scientific progress within the scope of this journal. It adds new ideas, applies reliable methods, and contributes with new data.

The manuscript is well organized, easy to follow and very well written. The introduction is very rich, referring global and regional previous contributions. Methods are very complete and correctly explained. The results are properly presented and the discussion is very rich. At some point, I consider that the authors could avoid the discussion about the changes in the species size as it is not even plotted in the main manuscript and later is not taken into account for the paleoceanographic implications discussion. Figures and tables are correct, even the Apendix ones. The references fit the journal requeriments. Below I detailed a few minor points, though mostly recommendations rather than criticisms. I recommend minor revisions and consider this manuscript fitting very well for the Journal Biogeosciences.

We thank the reviewer for their very positive comments.

Line 6: Is it "those" o "the"?

We changed it to "the".

Line: 23: I would write "e.g." because the sentence is refers to global reconstructions of past ocean conditions, and globally there are plenty more contributions than those cited in line 23.

Done

Line 43: The sentence is hard to understand. I suggest to shortened it in order to make the idea clearer.

The sentence was divided into two parts as below:

"However, this approach may be associated with uncertainties arising from a myriad of processes during the settling, deposition, and burial that may lead to varying degrees of proxy signal alteration (Regenberg et al., 2014). Furthermore, additional uncertainty stems from selected proxy calibrations and the instrumental database used for comparison with the proxy." @Line 45-49

Line 176: Try to avoid sentences that sound like "discussion" in the results. Lines 176 and 177 are an example of this.

The sentence was rephrased to:

"The multivariate analysis performed on hydrographic data separates the sites into two main groups, i.e., transects 1-3 in non-upwelling sector and transects 5-7 in upwelling sector." Line 185

Line 305: Please, check if the idea of this sentence is correctly expressed

The sentence has been rephrased to improve clarity.
Now It reads:
"Therefore, the inclusion of dead specimens may not necessarily result in a severe bias in the habitat depth estimates. Furthermore, the agreement in the habitat depth of *T. trilobus* inferred from sediments and our ALD calculation also suggests that dead specimens likely do not make up a large portion of the net samples; the same is probably true for *G. ruber* from the same samples. Together, these observations suggest that the relatively deep ALD calculated for *G. ruber* (white) is likely a robust finding and not severely biased by the inclusion of dead specimens in the calculation". @Line 334-338

**Reviewer #2**

General comments

The manuscript by Tapia et al. presented valuable planktic foraminifera dataset off Indonesia where the available data are currently limited. Their plankton net study along with published sediment trap and surface sediment records will form the basis for understanding planktic foraminifera ecology, seasonality, and foram-based paleoceanography in this important region. The manuscript deals well with the contrasting foraminifera distribution in upwelling sector and non-upwelling sector, and also includes implications for paleoceanography, which thus falls within the scope of BG.

However, the authors concentrates on six of 29 species, and do not show all of the species list. My concern is whether the taxonomic concept is consistent between plankton net, sediment trap, and surface sediments. The current form of the manuscript lacks the discussion of rare species. Another concern is the consideration of anthropogenic climate change. Plankton net and sediment trap are susceptible to recent climate change, though surface sediments likely hold pre-industrial state. I also felt that figures can be improved to better convey the results and discussion of the paper. I recommend major revisions of the manuscript.

We thank the reviewer for their positive comments and constructive criticisms. We have incorporated suggested changes and hope that these changes have improved the clarity of the manuscript.

Specific comments

1. A total of 29 species were identified, but all the species never appeared in the manuscript. The authors tend to discuss the major 6 species, but rare species also hold important information of water column structure and thus implication for paleoceanographic reconstruction. Although the manuscript compared the number of species between plankton net, sediment trap, and surface sediments (Discussion 4.1), is the taxonomic concept the same? If any difference exist, the authors should care the consistency to discuss the diversity of foraminifera assemblage. Also, taxonomic identification rely on the works before 1989. But the genus Trilobatus should follow the paper by Spezzaferri et al. (2015 PLOS ONE). I would like to see all the species of plankton net, sediment trap, and surface sediments to infer seasonality and possible dissolution effect on both major and minor species.

We do agree with Reviewer #2 that the full assemblage may be useful for other studies that are more focused on the total assemblage. We have included the complete assemblage data in the Supplementary Info for future consideration by the community. But we emphasize that the main focus of our study is not on the assemblages. Instead, we aim to (1) assess the habitat depth of species that are often used for geochemical analyses in paleoceanographic reconstructions (thus not assemblage), (2) compare the habitat depth of these species inferred from different sample types and approaches, (3) assess the implications for paleoceanographic reconstructions. To achieve these goals, we selected several species that are often used for geochemical analyses, for instance those used in sediment traps, surface sediment or downcore reconstructions (e.g., Mohtadi et al, 2007, 2009, 2011, 2014, 2017).

Adding more discussion on rare species and the full assemblage will not improve the clarity of the manuscript. The rare species are not suitable for the calculation of ALD as already outlined in the Methods section as a few specimens likely do not yield statistically significant estimates (Rebotim et al., 2017; Lessa et al., 2019).

As mentioned above, one of the main goals of this study is to compare the habitat depth inference from plankton net samples with the estimates based on surface sediments (Mohtadi et al., 2007)

and sediment traps (Mohtadi et al., 2009). Therefore, we used the same taxonomic concept as these previous works, and the lead author of those papers are in fact also contributing to this paper.

To clarify the above-mentioned point about taxonomic concept, we have added in the Methods section:

"We used the same taxonomic approach as in previous studies based on surface sediments (Mohtadi et al., 2007) and sediment trap (Mohtadi et al., 2009/11). The only exceptions are for G. elongatus and T. trilobus, as the names of these species have been updated recently by Aurahs et al. (2009) and Spezzaferi et al. (2015), respectively." @Line 106 -108.

2. Jonkers et al. (2019 Nature) paper presented modern plankton community driven by anthropogenic climate change. I'm wondering whether recent climate change affects plankton net and sediment trap data, which potentially alters the relationship of foraminifera assemblages between plankton net, sediment trap, and surface sediments. Coincidentally, Jonkers et al. paper includes one sediment trap data off Indonesia (Mohtadi et al., 2009) and categorizes apparent warming for this region (historical change is cooling but the species composition shows warming). What is the relationship between this study and Jonkers et al. paper?

We agree with Reviewer #2 that this is something interesting to add to the discussion. We have added some text to mention the reported warm bias in the assemblage in surface sediments due to anthropogenic effects and the likelihood that the water column may have changed over the last few decades, and discuss whether it has a bearing on our findings.

"A recent global compilation study which includes the sediment trap data from Indonesia reported a warm bias in the assemblage in surface sediments due to anthropogenic effects and the likelihood that the water column may have changed over the last few decades (Jonkers et al., 2019). We note that their approach is based on the biogeography of planktic foraminifera, i.e., each species occupies a specific thermal niche, which may span a temperature range of >10ºC for some tropical species. On the other hand, our main findings about the habitat depth and implications for paleoclimate reconstruction are based on individual species. The selected species are not dwelling at the limit of their thermal niche, thus as long as these species do not substantially shift their thermal niche over time, we do not expect any large bias due to the reported anthropogenic changes in foraminiferal assemblage." @Line 243-249.

3. The authors stated that Ujiié (1968) paper is the only study using plankton net off Indonesia (L58). Then the author's study is consistent well with the Ujiié paper? Currently, there is only a general description (L252-254), and no comparison of species found and its standing stocks. Even though the Ujiié paper did not investigate the vertical distribution, at least surface distribution of foraminifera should be discussed.

We agree with Reviewer #2 that this is something worth discussing. We did not discuss the results of Ujiie because (1) his samples were collected in a different season (winter, whereas ours were collected in summer), and (2) he only looked at foraminifera that are > 330 micrometer (whereas the bulk of foraminifera in our samples come from the size fraction <300 micrometer). Despite the aforementioned disparities in sampling, comparing Ujiie's data with ours may shed light on seasonal differences in assemblage off Java, and further allows a comparison with the assemblage in surface sediments. We will add this discussion in section 4.1.

"Interestingly, despite methodological differences (sampling season and water depth, size fraction analyzed), our results are broadly consistent with those of a plankton net study carried out here in late autumn-early winter of 1963 at the end of the upwelling season (Ujiié, 1968). Ujiié found that the assemblage of planktic foraminifera off Java was consisted of a mixture of species associated with nutrient-rich and nutrient-poor waters, dominated by *N. dutertrei* (28%), *G. ruber* (22%) and

*T. trilobus* (10%). On the other hand, off Sumatra (100º E) oligotrophic species *T. trilobus* and *G. ruber* accounted for 56% of the total assemblage of planktic foraminifera. Thus, the assemblage and dominant species characterizing these two sectors seem to persist until the end of the upwelling season." @Line 270-276

4. Based on Figure 9, thermal gradient of plankton net in non-upwelling sector is 2 degrees C. However, the thermal gradient (delta T) seems much larger in the same sector in Figure 10. I couldn't follow the apparent difference in delta T between Figures 9 and 10. Please show absolute values of water depth and temperature in Figure 10, rather than relative values.

We thank the reviewer for pointing out this confusion. The temperatures plotted in Figure 9 are abundance-weighted temperatures (i.e., more weight is given to the depth interval with higher abundance). For Figure 10 we simply marked the ALD of selected species along the measured temperature profile without any calculation.

We have added detail of the calculation in the caption of Figure 9, and provide absolute values of water depth and temperature in Figure 10.

5. There are two discrepancies between plankton net data and surface sediment records. One is average living depths in Java-LSI (Figure 8). The other is thermal gradient in Sumatra and Java-LSI (Figures 9 and 10). What is the exact relationship between two discrepancies? If the discrepancy of the average living depths in Java-LSI is resolved, then the other discrepancy is also resolved?

Plankton net data off Java suggest a much deeper ALD compared to the calcification depth inferred from geochemical data of foraminifera in surface sediments. Consequently, the abundance-weighted temperatures based on the plankton net mixed-layer species (red boxplot in the bottom right panel in Figure 9) with a greater-than-expected habitat depth are also lower than that suggested by surface sediment data (red boxplot bottom left panel in Figure 9). In other words, the discrepancy in the thermal gradient is due to the different calcification temperature / habitat depth of the mixed layer species in Java-LSI.

To improve clarity, we have outlined the temperature calculation in the caption of Figure 9 to emphasize the difference in estimating the depth, i.e. geochemistry vs. observation.

We also added several sentences throughout the discussion to improve clarity:

"The habitat depth change of the mixed-layer species is thus the primary reason for the $\Delta$T difference between the two sectors." @Line 441

"Plankton net data suggest a greater habitat depth for the mixed-layer species and hence also lower inferred temperature. As a result, the $\Delta$T off Java-LSI calculated from the plankton net data is smaller than that of surface sediment data." @Line 458

Although the authors already pointed out the different temporal coverage of sample types, as the Referee (and as a reader), I expect the authors to discuss possible solution for the discrepancy. Please consider the above comments (specific comments 1 to 5) to utilize valuable dataset to tackle the discrepancies between plankton net and surface sediments (and sediment trap).

We find this suggestion very useful and have added a few suggestions for future work at the end of the discussion:

"To further shed light on the transfer of proxy signal from the water column to the sediment, longer sediment trap time series and repeated plankton net sampling in the same region will be useful to capture the seasonality of the vertical distribution of planktic foraminifera. Importantly, generating geochemical data on plankton net samples may help to verify the habitat depths and allow a direct comparison with the depth inference from the surface sediments. It would also be helpful to constrain the age of surface sediments to ensure that they are comparable to modern data." @Line 481-485.

Technical corrections

The manuscript uses the Ocean Data View and R software to plot and analyze the data. But no references and acknowledgements is presented. Please appropriately refer the ODV and R software.

References added.

L21 In addition to Katz et al., 2010, add seminal paper.

We added the seminal papers from Bemis et al. (1998) and Fairbanks et al. (1980).

L28 Abbreviation of SST should be in L27.

Changed as suggested.

L29 For transfer function, add seminal paper (e.g., Imbrie and Kipp, 1971).

Added.

L34 Add oxygen isotope before "d18O", and add ratio after "Mg/Ca".

Added.

L35 Rephase "popular".

Replaced with "powerful".

L63 Add period after the end of sentence.

Done.

L104 Both sensu stricto (s.s.) and sensu lato (s.l.) are not italic. See Wang (2000) paper.

Done.

L111 What is the approach of Mohtadi et al. (2009) to differentiate N. dutertrei from N. incompta? Please explain briefly.

Brief description of the approach in method section.

"…based on the presence of an umbilical tooth, and the occurrence of more than four chambers per whorl."

L140 Delete "psu". No unit for salinity.

Done.

L173 Delete "sea".

Done.

L184-L185 It is not clear that off Sumatra means transect 1-3, and Southern Sumatra and Java-LSI mean transect 4-7. Please clearly state which transect you mention, instead of area's name.

The transects were added in the text.

L196-L197 I'm not sure these references for what reasons. Six species have been often used in paleoceanographic studies? Then describe so.

We rephrased the sentence to improve clarity.

"In the following section, we describe the vertical distribution of six species of planktic foraminifera that are typically used in paleoceanographic studies (for example, Caley et al., 2012; Ding et al., 2013; Mohtadi et al., 2017; Steinke et al., 2014; Tapia et al., 2019), namely *T. trilobus, G. ruber* (white), *G. bulloides, N. dutertrei, P. obliquiloculata*, and *G. menardii.*" @Line 204–206

L205 Typo, lysocline.

Corrected.

L208 Not Fig. 5g, but Fig. 5f.

Corrected.

L211 Not Fig. 5f, but Fig. 5g.

Corrected.

L217 Delete ")" after G. menardii.

Done.

L228 Any reference for the lysocline depth?

Two references have been added (Ding et al., 2006; Mohtadi et al., 2007)

L260 and L381 Change from planktonic to planktic.

Done.

 L263 Add "(white)" after G. ruber.

Done.

L296 How to calculate habitat depth from surface sediments? Please explain.

We added additional explanation in the Introduction outlining the approach used to obtain habitat depth estimate from surface sediments.

"In this approach, habitat depth is defined as the water depth at which the reconstructed Mg/Ca-temperature or seawater $\delta^{18}O$ value show the closest match with the instrumental data or climatological product." @Line 43-46

L303 Delete "inclusion of".

Done.

L308 Sort species name as in L286. Be consistent with the species order.

We changed the order of L286 so that it is consistent with the order of the species appearing in the discussion.

L332 Geochemical data of planktic foraminifera? It is not clear. Also, what is c of d18Oc? Calcite? State clearly.

We spelled out "Calcite".

L346 Rephrase "greater" to deeper.

We would like to keep "greater" because we find "deeper depth" a tad awkward grammatically.

L380 Delete "Possible". Implication itself includes possibility.

Done.

L403 It is not clear the meaning of thermal gradient "of" mixed-layer and deep-dwelling species. Perhaps thermal gradient "between" mixed-layer and deep-dwelling species?

Modified as suggested.

L427 Does parentheses need for delta T?

Parentheses were removed.

L466 Typo, LSI.

Corrected.

Figure 1. Add the island names (Sumatra, Java, and the LSI) to Fig. 1a. Some readers are not familiar with this region.

Modified as suggested.

Figure 2. I suggest to add horizontal lines (like error bars) on top of Fig. 2a showing each transect (1 to 7) corresponds what longitudes. In other words, 7 horizontal lines show longitudinal

extent of each transect, which helps readers to understand regional contrast of temperature, salinity and so on. This is also true for Figure 5.

We have added horizontal lines in Figure 2 to indicate the longitudinal extent of each transect.

Figure 4. Add explanation for box plot. What is the meaning of box and bars? For stacked graph, legend is ascending order but the actual data is presented as descending order. I prefer ascending order also for the data.

We added a schematic to explain the boxplots. The panel (c) was changed to match the ascending order of the legend.

Figure 5. The figure is currently shown up to 600 m. But the maximum water depth should be 500 m. Please limit the water depth.

We changed the axis limit to 500 m.

Figure 6. Similar to Figure 4, add explanation for box plot. Are the axis logarithmic? It is not clear, since no axis is shown between 100 and 200 m depth. Also, significant digits should be the same (1 or 2?) for the median of ALD. Add space between "species" and "T. trilobus" in the figure caption.

We added a schematic explaining the boxplot, included y-axis marks for 100-200 m and changed the significant digit to 1, and added the missing space.

Figure 8. Increase the font size of the species name. Remove italic from (white) for G. ruber.

Done.

Figure 9. Add legend for red and blue colors, instead of stating in the figure caption.

Modified as suggested

Figure10. In the figure caption, there are a and b. But a and b are not present in the figure. Be consistent with the caption. I prefer absolute values of water depth and temperature, rather than relative values.

We labeled the panels and added absolute values of water depth and temperature.

Table 1. Put space after period for P. obliquiloqulata.

Done.

Table A1. Add "dd.mm.yy" for Date.

Done.

---

## Referee Report (RR1)

The study by Tapia et al. showed a new and relevant planktic foraminifera dataset off Indonesia, and presented the implications of their results to paleoceanographic works using foraminiferal proxies. The manuscript is very well written and fits the scope of the journal. I recommend the acceptance of this manuscript after very minor revisions.

Major comments:

1. The authors need to distinguish habitat depth and calcification depth. They should write some sentences to explain the difference between these two. Also, when comparing their plankton net results with the surface sediment samples, they should mention that the surface sediment samples provide an estimation of calcification depth and not habitat depth. Maybe some of the discrepancies between plankton net and surface sediment samples reported by the authors are related to this issue.
2. L303: The authors should discuss a little bit more their findings regarding synchronized reproduction. How they compare with other studies? The authors show only results for the size, but not for the abundance of species before and after the full moon. I suggest that they discuss more about synchronized reproduction, showing the abundance data before and after the full moon, and comparing the findings with other studies.

Minor comments:

L163: The authors describe how they have defined the thermal mixed layer depth. The approach used to define the MLD is different from the one of Boyer Montégut et al. (2004). I suggest that they have a look at this work to see if there is any difference in MLD estimations using different approaches.

de Boyer Montégut, C., Madec, G., Fischer, A. S., Lazar, A., and Iudicone, D. (2004), Mixed layer depth over the global ocean: An examination of profile data and a profile-based climatology, J. Geophys. Res., 109, C12003, doi:10.1029/2004JC002378.

L203: Close the parentheses in the sentence

---

## Author Response (AR2)

**Institute of Oceanography**
**National Taiwan University**

**June 10, 2022**

**Response Letter Manuscript bg-2021-329**

Dear Dr Kitazato,

Here, we would like to re-submit the revised version of the manuscript titled "Contrasting vertical distributions of recent planktic foraminifera off Indonesia during the southeast monsoon: implications for paleoceanographic reconstructions"

We would like to thank the editor and the anonymous reviewers for their helpful comments and suggestions. We have made changes to our manuscript accordingly by adding some text in order to clarify some concepts considered unclear by the reviewers. We have modified the Figure 4 in order to better visualize our results. Additionally, we have corrected some values in the Table A1 as well as the visualization of the data in the Figure 3.

A list of all the changes made can be found in the point-by-point response to the reviewers' comments. In addition, we have annexed submit two versions of the revised manuscript, i.e., with tracked changes. We refer to the line number in our rebuttal letter as viewed with tracked changes. For your convenience, the reviewers' comments are in black while our responses are in blue.

We hope that we have successfully addressed all the points raised and that the improved manuscript now meets the standards of Biogeosciences.

Thank you very much for handling this manuscript.

Yours sincerely,
Raúl Tapia

No. 1, Sec. 4 Roosevelt Road | Tapei | Taiwan (R.O.C) | Phone: +886-2-33661386 | Email: raultapia@ntu.edu.tw

[Figure]

[Figure]

Referee #2

The author's addressed appropriately my previous comments, and the clarity of the manuscript improved from the initial version. I have several but only minor comments below (Line numbers refer the tracked changes file.). I recommend technical corrections of the manuscript.

L86 LSI is already abbreviated in L63. Delete "Lesser Sunda Islands" here.

Done

L91 Not SO184, but SO-184.

Done

L192-L194 It would be better if some information on transect 4 is added. According to Figure 4, high median value of abundance (~35) at transect 4 perhaps reflects highest abundance (~80) at Station 10048.

As correctly pointed out by the referee, extremely high or low values may bias the mean value of a distribution of observations. Because of this we decided to describe the transects using the median values (instead of mean) which are less sensitive to the effect of outliers. The median value obtained for Transect 4, i.e., ~35, is from Station 10036. Also, we note that Station 10048 is in fact in Transect 5 and not 4.

To avoid this confusion, we have added the mean value of each transect in Fig 4 in addition to the median value to emphasize their difference. We have also added in the caption of Fig 4 that "White dots denote the mean of data distribution. Note the difference between the mean and median, where the former is more susceptible to extreme low/high values."

L511 I found all the species list (n=29) of planktic foraminifera in PANGAEA database. Please make it clearer what kind of data you stored in the database (not just "data generated in this study" but like "planktic foraminifera species and abundances in this study"). Also, somehow the diatom species "Triceratium barbadense" appears in the database webpage (#9 as counting, foraminifera, planktic). Please check again the data you stored, to assure your research.

We appreciate the Referee's thorough examination. We have contacted Pangaea to fix the discrepancy between what is being displayed on Pangaea's webpage and the file submitted by us (same file as the one added as supplementary information to the manuscript). The column number 9 labeled as "Triceratium barbadense (%)" was corrected to its original label "Water filtered (m^3)".

L673, L692, and L698 Delete some numbers (70, 78, and 81, respectively) at the beginning of the added references.

Done

L677 Superscript for d18O and d13C.

[Figure]

[Figure]

Done

Figure 6 Better to rewrite the blue sentence in the figure caption. There are many subjects in one sentence.

The caption was rephrased to "Schematic of the box plots showing the median value (horizontal line) and the whisker marks the minimum (min) and maximum (max) values. Symbols (white circles) depict the ALD values."

[Figure]

[Figure]

Referee # 3

The study by Tapia et al. showed a new and relevant planktic foraminifera dataset off Indonesia, and presented the implications of their results to paleoceanographic works using foraminiferal proxies. The manuscript is very well written and fits the scope of the journal. I recommend the acceptance of this manuscript after very minor revisions.

We thank the Referee for their favorable review.

Major comments:

1. The authors need to distinguish habitat depth and calcification depth. They should write some sentences to explain the difference between these two. Also, when comparing their plankton net results with the surface sediment samples, they should mention that the surface sediment samples provide an estimation of calcification depth and not habitat depth. Maybe some of the discrepancies between plankton net and surface sediment samples reported by the authors are related to this issue.

We agree and have included some text in the introduction L41–46 and Discussion Section 4.6 L464 clarifying the difference between habitat depth and calcification depth.

"One common way of inferring planktic foraminifera habitat depth is by using the calcification depth, obtained by comparing the reconstructed parameters (typically Mg/Ca-SST) from surface sediments with instrumental data or climatological products (e.g., World Ocean Atlas) (Groeneveld and Chiessi, 2011; Hollstein et al., 2017; Mohtadi et al., 2011; Steinke et al., 2014; Tapia et al., 2015). In this approach, calcification depth is defined as the water depth at which the reconstructed Mg/Ca-temperature or seawater δ18O value shows the closest match with the instrumental data or climatological product. However, calcification depth does not necessarily coincide with the inferred habitat depth where the organisms are observed during sampling. Also,... "

"This discrepancy between surface sediment and plankton net data off Java may stem from the different temporal intervals integrated by each sample type and the fact that calcification depth inferred from surface sediment may not be synonymous with the habitat depth inferred from plankton net data."

2. L303: The authors should discuss a little bit more their findings regarding synchronized reproduction. How they compare with other studies? The authors show only results for the size, but not for the abundance of species before and after the full moon. I suggest that they discuss more about synchronized reproduction, showing the abundance data before and after the full moon, and comparing the findings with other studies.

We refrain of discussing synchronized reproduction based on the abundance of species due to a limitation in our methodology. Normally, repeated sampling before and after the full moon in the same region/site is required (e.g. Meilland et al., 2021). In our case, however, the sampling strategy does not allow us to isolate the effect of the location and reproduction in the abundance parameter as the samples collected before the full moon are located off Sumatra (low abundance) while the samples collected after full moon are located off Java (high abundance).

We have added a sentence at L308 to clarify this:

"We note that samples off Sumatra (Java-LSI) were collected before (after) the full moon, making it difficult to disentangle the effect of hydrography and synchronized reproduction."

Minor comments:

L163: The authors describe how they have defined the thermal mixed layer depth. The approach used to define the MLD is different from the one of Boyer Montégut et al. (2004). I suggest that they have a look at this work to see if there is any difference in MLD estimations using different approaches. de Boyer Montégut, C., Madec, G., Fischer, A. S., Lazar, A., and Iudicone, D. (2004), Mixed layer depth over the global ocean: An examination of profile data and a profile-based climatology, J. Geophys. Res., 109, C12003, doi:10.1029/2004JC002378.

We favored the use of a regional definition of MLD over the global MLD definition (ΔT > 0.2 ºC) proposed by Boyer Montégut et al. due to the following reasons. The number of Temperature-Salinity profiles off Indonesia considered in their analysis is < 10 and mostly collected from more pelagic settings (Figure 1b in Boyer Montégut), therefore there is an underrepresentation of the more coastal Indonesian area where our stations are located. Although the use of a global definition of MLD on our dataset would lead to a reduction in the thickness of the MLD, it has no effect on the zonal pattern, i.e., there is a thicker MLD off Sumatra and thinner MLD off Java.

We have now added a sentence in the manuscript at L165 to clarify this:
"We note that a different approach in defining mixed layer depth (e.g., Boyer Montégut et al., (2004) for global ocean) would result in different thickness of the mixed layer but this does not change the spatial pattern that is the focus of the discussion here."

L203: Close the parentheses in the sentence
Done

[revised manuscript text omitted]